# Physical Fitness as a Predictor of Performance during Competition in Professional Women’s Basketball Players

**DOI:** 10.3390/ijerph20020988

**Published:** 2023-01-05

**Authors:** Sergio J. Ibáñez, María Isabel Piñar, David García, David Mancha-Triguero

**Affiliations:** 1Group for Optimisation of Training and Sport Performance (GOERD), Faculty of Sport Science, University of Extremadura, 06006 Badajoz, Spain; 2Faculty of Sport Science, University of Granada, 18011 Granada, Spain; 3Department of Physical Education and Sport, Cardenal Spínola, CEU San Pablo Andalucía University, 41930 Sevilla, Spain

**Keywords:** woman, inertial device, game indicators, physical fitness, match analysis, notational analysis, predictive analysis, game position

## Abstract

The evaluation of physical fitness in team sports is enjoying greater importance in the training of professional teams. The objectives of this research were to characterize physical fitness and game indicators based on the game position. This is an empirical study, with a quantitative, descriptive and cross-sectional methodology. In addition, different relationships between the level of physical fitness and the game indicators during the competition were determined. Finally, a predictive analysis of the selected variables was carried out in order to know the importance of the variables in the performance and at what time of the season they had the greatest impact. For this, a professional female basketball team (*n* = 12) with a mean age of 25.25 ± 7.617 years, height 178.25 ± 9.206 cm and a body mass of 72.33 ± 11.657 kg was analyzed. Each player was equipped with a WIMUPRO inertial device, and all competition statistics were analyzed. The results obtained show that there were no differences among all the skills evaluated and game indicators depending on the game position. Likewise, a relationship was determined between the level of physical fitness and the technical–tactical contribution, being different between different times of the season. Moreover, physical fitness only predicted the player’s final performance in the competition for female player centers in the first and second rounds of the championship, and for forwards in the first round. Four physical-physiological profiles with contributions to the different ones during the competition were also determined. Finally, three groups of female players by playing position were identified according to their sport performance, namely PIR, high, medium and low ratings, associated with physical performances. In all groups, there were significant differences between playing positions, PIR and physical performances.

## 1. Introduction

Basketball is one of the most practiced sports worldwide. This social repercussion also coincides with a high production of scientific documents, being one of the sports in which more research is carried out [1]. These investigations have different objects of study such as psychological aspects [2,3], nutritional aspects [4,5,6], technicians and tactics [7,8,9], physical fitness (PF) [10,11] or game indicators (GI) [12,13,14]. Research approaches are sometimes conducted in isolation, without interaction between the different topics. The evolution of basketball research requires interdisciplinary studies in which more than one object of study is related, as this sport is complex, in which multiple variables interact. This research aims to identify the relationship between two of them, analyzing the relationship that may exist between physical fitness and game indicators.

A review of the literature shows that a large majority of articles that focus their object of study on the GI are carried out in men’s basketball [15,16,17,18]. Proof of this is evidenced by Ibáñez, et al. [19] who demonstrated the importance of certain GI in the final result of the competition and determined that the team with the highest number of actions in the defensive phase (blocks and rebounds) and two-point shots (in the attack phase) was more likely to achieve victory [20]. Regarding the GI, in the literature there are documents that relate the GI with the playing position [1,8,16,19]. In this line, Sampaio, Ibáñez, Gomez, Lorenzo and Ortega [15] and Escudero-Tena et al. [21] determined that point guards and shooting guards contributed to the team a greater number of steals, blocks, fouls committed, shots and three points and assists, while the centers were characterized by providing the team with rebounds, blocks and two-point shots. Specifically in women’s basketball, Fernández-Cortes et al. [22] identified that the players did not have the same contribution to the game, depending on their game position, as the phase of the competition. While the point guards make a greater contribution in the field goals and in the two-point shots, the power forwards contribute more free throws during the regular league. During the play-offs, point guards and power forwards increase their effectiveness on three-point shots. During the play-offs, the offensive game is favored by the contribution of the guards through assists, while the defensive game is increased by the contribution of more rebounds by the centers and the defensive intensity through steals by the forward players [23].

Additionally, GI can be affected based on different contextual variables. In this line, three types of matches were described based on the results, namely equal, unbalanced and very unbalanced), according to the classification of Ibáñez, et al. [24]. Using this classification and the analysis of the GI, Gómez-Ruano, Lorenzo, Sampaio, Ibáñez and Ortega [23] stated that even games are decided by the number of rebounds, while lopsided games are decided by scoring on two-point shots. In addition, Gómez, Lorenzo, Ortga, Sampaio and Ibáñez [16] confirmed that the contribution to the team and the GI obtained by the starting and substitute players are different. The authors of [25] stated that depending on the game position, players presented differences in physical and technical–tactical aspects. These requirements were related to the actions performed by each position within the game [26].

The variety of players’ GI contribution depending on the game position has an impact on the demands and requirements that they support in the competition [27]. These results may be due to the differences in the anthropometric characteristics of the players, linked to the playing position [28]. For this, one of the most used methods to know the physical-physiological state of the athlete is the evaluation of the PF. This PF evaluation can be carried out through PF tests [29] or through the analysis of the demands and requirements of the athlete in competition [30]. Regarding the differences, the PF tests evaluate an isolated ability or capacity [31], while evaluating the competition can provide quality information, but it must take into account aspects related to the rival, the scoreboard or other variables that could affect the athlete [30]. The importance of knowing the state of PF of the athlete allows individualizing loads, confronting the athlete with planned stimuli, and designing a plan where the desired characteristics are improved [32,33]. Among the main differences, it is differentiated by sex, since the load supported both in the PF test and in competition between men and women is different [34]. In addition, the assimilation and absorption of the load also varies, which may affect aspects of training or injury prevention [35].

To recruit players for a team, coaches use various objective and subjective information. Among the objective information, they use the Performance Index Rating (PIR), which is provided by the official competitions and is public [36]. This information can be classified according to the specific position. There is little public information on the physical performance indicators of the players that would allow an adequate selection of players according to the needs of the game. Only in the NBA are rookie players given physical and technical tests, the Draft Combine. The results of these tests are not always directly related to subsequent performance during their first year of competition [37].

To the best of our knowledge, there are few studies that analyze the influence of PF on GI in women’s basketball [13,16,20], since the vast majority of research focuses on men’s basketball. Therefore, the general objective of this research was to know the influence of the results of the physical fitness tests on the game indicators of professional basketball players. To develop this objective, four specific objectives were proposed: (i) to analyze the differences in the results of the PF tests depending on the game position; (ii) to identify the differences in the GI of the players depending on the game position; (iii) to identify the relationships between PF and GI depending on the moment of the season; (iv) to predict the relationship between the results of the PF tests and the performance of the players depending on the moment of the season.

Based on these objectives, the following hypotheses are made: (a) there will be differences in physical fitness results among female players according to their specific position; (b) there will be differences in game indicators among female players of different specific positions; (c) during each moment of the season, there will be specific relationships between physical fitness and game indicators; (d) there will be some physical fitness test results that can predict the performance of female players.

## 2. Materials and Methods

### 2.1. Design

The proposed research is classified as a methodology of quasi-experimental studies, because they focus on the analysis of the differences depending on the position of the game. In addition, the group design is equivalent through the study of the PF and the results of GI obtained in the competition [38]. Specifically, this is an empirical study, with a quantitative, descriptive and cross-sectional methodology [39].

### 2.2. Participants

Convenience sampling was used, as it is a non-probabilistic and non-random sampling technique. This sample was selected according to the ease of access offered by the club and the availability of the female players to be part of the sample. The researchers selected this sample because the team was already participating in other research on the quantification of training and competition loads. The head coach was contacted first to find out his availability. After agreeing to participate in the study, permission was sought from the club.

A professional women’s basketball team, which participates in the Spanish Liga Femenina Callange, participated in this study. The team was composed of 13 female players. The reality of sports training is that the number of players analyzed is always relatively low, since it is a high competition context. This reality of the high-level sports context does not limit the validity of its results [40]. Finally, only 12 female players were included in the study as they met the inclusion criteria.

All team members (players) underwent PF assessment tests. In addition, prior to carrying out the PF assessment, all the players and coaching staff were informed about the protocol to be carried out, as well as their queries or clarifying doubts were answered. Subsequently, all the members signed the informed consent, and the research was carried out following the ethical criteria of the Declaration of Helsinki (2013) and approved by the Bioethics Committee of the University (233/2019).

#### Eligibility Criteria

The following criteria were used to select the cases participating in the sample: (i) only female basketball players belonging to the club analyzed were included in this study; (ii) to be considered a case study, the female player must have participated for at least 12 min per game [41]; (iii) the player must have played for at least 1 year in the professional category; (iv) the player must have performed the physical fitness tests during the pre-season; (v) the player must have completed the two weeks of previous training, (vi) the player must not have suffered injuries in the first weeks of training.

Exclusion criteria were: (i) participating less than 12 min of play in each match analyzed; (ii) not having actively participated in the physical fitness test, (iii) being involved in another type of sport at the same time as in this study; (iv) not having a sufficient cognitive level to collaborate in the study or not being able to attend the established sessions.

### 2.3. Sample

The sample consisted of a total of 230 cases (114 cases in the first round and 116 in the second round of the championship). Each case represents the participation of a player who has played at least 12 min per game.

The data were collected during the 2020–2021 season. Physical tests were conducted at the end of the pre-season macrocycle. Data on game indicators were recorded during the 26 games played during the first and second rounds. The season was structured in eight macrocycles, one of which lasted five weeks and seven of which lasted four weeks. (Figure 1). The start of the preseason was during the month of September.

### 2.4. Variables

The game position (guard, forward and center) in this investigation was used as an independent variable. The rest of the variables were grouped according to their origin (PF, performance variables or competitive game variables).

The rating of the players, evaluated by Performance Index Rating (PIR) was the dependent variable for the prediction study. The PIR is calculated with the following formula from the game indicator [36]:PIR = (Points + Rebounds + Assists + Steals + Blocks + Fouls Received) −(Missed Field Shots + Missed Free Throws + Turnovers + Received Blocks +Committed Fouls)

#### 2.4.1. Physical Fitness Performance Variables

The selection of the variables used in this research was carried out based on research with a common theme [10,11,30]: (i) height of jumps (Height); (ii) total time of the 5 sprints (Total Time); (iii) average time of the 5 sprints (Avg. Time); (iv) time spent to perform the test; (v) explosive distance (Expl. Dist.); (vi) maximum speed (Max. Speed); (vii) distance; (viii) relative heart rate (%HR Max); (ix) Player Load (PL).

The physical fitness variables were obtained after performing the SBAFIT test battery [25] at the end of the preseason period. The precise description of all the physical fitness tests, as well as the process of obtaining the variables for each test, was carried out following the instructions defined by Mancha Triguero, García-Rubio and Ibáñez [29].

#### 2.4.2. Competitive Game Variables

These variables were obtained from the game indicators (GI) provided by the competition itself on each of the official matches played by the team (1 full season, 28 official matches). Among the most notable variables are:

(i) minutes; (ii) total points; (iii) efficacy two-point shots (scored/thrown); (iv) efficacy three-point shots (scored/thrown); (v) free throws (scored/thrown); (vi) defensive rebounds; (vii) offensive rebounds; (viii) total rebounds; (ix) fouls committed; (x) fouls received; (xi) assists; (xii) blocks; (xiii) steals; (xiv) turnovers; (xv) valuation (Performance Index Rating, PIR). These variables were chosen because they are the most used in research with similar themes [1,13,19].

### 2.5. Materials and Instruments

#### 2.5.1. Materials

To carry out the research, different materials were used depending on the type of variable. For the PF variables, a WIMUPRO^TM^ inertial device (RealTrack Systems, Almería, Spain) was used to obtain the external load variables and a GARMIN^®^ heart rate band (Olathe, KS, USA) for the internal load variables. In addition, an Ultra-Wide-Band (UWB) technology system [42] was used to obtain a Local Positioning System (LPS) made up of 8 antennas, which provides higher quality and reliability for the results obtained [43]. The reliability and validity of these inertial devices has already been shown for research in sports, and specifically to analyze the internal and external load of the athlete [44,45,46].

#### 2.5.2. Instruments

The PF tests to which the players underwent were part of the SBAFIT Battery [29]. The players underwent the full battery and performed the tests:

(i) Abalakov test [47], (ii) multi-jump test; (iii) right/left centripetal force; (iv) SIG/ANA anaerobic test [48]; (v) SIG/AER aerobic test [48]; (vi) Repeat Sprint Ability, RSA, test 5 × 14 m [29]; (vii) *T* Test with ball/without ball [29].

Finally, to obtain the GI of the competition, the box scores provided by the organizer of the competition at the end of the match were obtained. This information is available on the competition’s official website (www.feb.es (accessed on 1 May 2021)).

### 2.6. Procedure

First, the coaching staff, sports management and players of the selected team were contacted. In a first meeting, the procedure was reported, and the protocol to be carried out and the tests that they were going to face were presented. Once the proposal was accepted by all the members, the dates were selected to carry out the PF evaluations. Prior to the evaluation, the players contacted the selected tests and the material they were going to wear in several training sessions. The tests were divided into two days separated by at least 72 h and the SBAFIT battery protocol designed by Mancha Triguero, García-Rubio and Ibáñez [29] was followed. The website of the organizer of the competition was accessed and they were obtained from all the matches. The evaluation of the PF was carried out in the last week of preseason (prior to the beginning of the season). This time was chosen because all the players had been training for some time, no player was injured and the PF was optimal.

### 2.7. Statistical Analysis

First, a descriptive analysis of PF variables (mean and standard deviation) was performed. Next, the criteria assumption tests were carried out (normality test) [49]. For PF variables, the *Shapiro–Wilk* test was used as it involved a sample of less than 50 cases (only 9 players met the requirements of playing at least 12 min per game). For the GI variables, the *Kolmogorov–Smirnov* test was performed with the *Lilliefors correction* as it was a sample of more than 50 cases. The results obtained showed a normal distribution of the data, so parametric tests were performed to test the hypothesis in the variables related to PF and no normal distribution of the data, so no parametric tests were performed to test the hypothesis in the variables related to the GI (Table 1).

To find out the differences between positions regarding the PF tests, an *ANOVA* test with *Bonferroni Post Hoc* was performed [50]. Furthermore, *partial eta squared* (*η*^2^) measures were calculated, namely, for a low effect (0.01–0.06), moderate effect (0.061–0.14) and high effect (>0.14) [51]. To find the differences in the GI variables normalized to minutes according to the specific position, the Kruskal–Wallis H test was used [49].

In addition, an analysis of bivariate correlations of the GI variables and differences between positions was carried out according to the minutes of play of each player, relativized per minute, (without taking into account the game position of the athletes analyzed). For this, the Spearman’s Rho correlation coefficient was calculated with the aim of looking for relationships between the PF tests and the performance obtained in the competition, because the GIs had a non-normal distribution.

A TwoStep Cluster was performed to identify the most important players based on the variables of minutes played and rating. To graphically present the differences between the groups of players, the results were transformed into Z-scores and then normalized to a scale of 0–100 for visualization and comparison.

To carry out a more precise predictive analysis, the database was segmented according to the game position and the moment of the season. Finally, a linear regression was carried out to predict the relationship between the results of the PF tests and the performance of the players, measured through the PIR based on the moment of the season.

All these calculations were performed with the statistical software SPSS 24.0 (SPSS Inc., Chicago, IL, USA), with the reference statistical value being *p* value < 0.05 [50].

## 3. Results

Sociodemographic and body composition characteristics are presented in Table 2 of the team, as well as by playing position.

### 3.1. Differences between PF According to Game Position

The results of Table 3 show that there are significant differences in all the PF tests carried out depending on the playing position in the Max. Speed of *T* Test without ball. In the rest of the variables, although there are differences in the descriptive results, there are no significant differences between game positions. The players with guards and forward positions present better results in all the tests than the center players. The main differences in physical fitness between positions may be related to anthropometric aspects and specific training. In addition, most of the variables have a very high effect of statistical power, giving greater validity to the data obtained. Secondly, the variables Height (Abalakov and multi-jump), explosive distance (centripetal force right and aerobic capacity), maximum speed (*T* Test without ball), Player Load (anaerobic capacity) present low or moderate effect size values.

### 3.2. Differences in GIs According to Game Position

The results of Table 4 show that there are significant differences in GI relativized per minute carried out depending on the playing position. The results obtained show the game position has an influence on certain game indicators. The game position of the players has an impact on the development of the competition, providing GI that are different from those that can be provided by other positions.

### 3.3. Relationships between PF and GI Depending on the Moment of the Season

Table 5 shows the results of the correlations between the FP and the IG. Only the variables in which at least one relationship was found are presented. The relationships in which they coincide in the first and the second round have been marked in green, while those that only occur in one round have been marked in orange. Most of the variables present constant correlations (in the first and second rounds of the championship). However, there are variables that are only correlated at one time. These variables are mostly related to effectiveness (two points, three points and free throws).

The results shown in Table 5 (Table 5a,b) indicate that multi-jump height, sprint speed, aerobic and anaerobic capacity are the skills that have the greatest relationship with game indicators. Regarding the game indicators, it is observed that the evaluation of the competition is the indicator that has the greatest number of relationships with PF, followed by the two- and three-point shots and rebounds. On the other hand, it is observed that the relationships between GI and PF variables undergoes an alteration between the first and second rounds, causing some relationships to disappear and new ones to form. Finally, it is observed that the Abalakov jump height variable (first and second round) and the duration time of the test and max. speed *T* Test without ball (first round) do not correlate with any game indicators variable.

### 3.4. Distribution of Female Players by Playing Position Based on Their Relevance in the Game

The cluster analysis identified a good model with two groups of players. The first group consisted of 49.1% of the sample and will be referred to as less important players (13.50 player minutes and 1.28 PIR). The second group consisted of 50.9% of the sample that will be referred to as important players (26.02 player minutes and 11.71 PIR). The results showed an equitable distribution among the guard players (43.4% of unimportant players vs. 56.6% of important players), unbalanced in the forward players (72% of unimportant players vs. 28% of important players), and again unbalanced in the center players (32.5% of unimportant players versus 67.5% of important players). In all clusters the differences between the generated groups were significant (*p* < 0.001).

### 3.5. Distribution of Female Players by Playing Position, Their Evaluation and Physical Fitness Test Results

Subsequently, a more specific cluster analysis was performed in which we tried to identify groups of female players by playing positions including the PIR and the nineteen physical performance variables. Three good cluster models were identified for each specific position, in which the most important predictor was PIR and with equivalent sizes (Table 6).

Specific profiles were found within each playing position for each group of female players according to their sport performance, PIR. Figure 2 shows the physical performance profiles for each player cluster by specific position.

The female players with the highest rating during the game are the female players who obtained the best results in the ANA test (distance, explosive distance, relative HR and Player Load), besides being the best in the distance and explosive distance of the AER test, and explosive distance of the CFL test. On the other hand, the female players with the highest scores in the PIR obtained the best results in the ABK jump height, covered more distance and explosive distance in the AER test, as well as had more Distance and Relative HR in the ANA test. Finally, the highest rated female players were the ones who had the best results in time and maximum speed in the *T* Test with and without the ball, they covered more explosive distance and explosive distance in the AER test, as well as more explosive distance in the ANA test.

### 3.6. Predict the Relationship between PF and Player Performance Based on the Time of the Season

Finally, Table 7 shows the results of the prediction model of a high PIR rating related to the results of the physical fitness tests. For the linear regression, we used PIR as the dependent variable and as independent variables one for each physical fitness test (Height ABK, Between Jumps MJ, Total Time RSA, explosive distance CF right; explosive distance CF left; Time *T* Test without ball; Time *T* Test with ball Player Load AER test; Player Load ANA test). The table shows the existence of six regressions, one for each playing position and time of the season. Only three of them are statistically significant.

The R^2^ value is high for female centers players and low for female forward players during the first round, explaining between 49% and 54% of the PIR for center players and only 25% of the PIR for forward players. The pivots between 46 and 61% of the PIR rating cannot be explained solely by the physical fitness in each of the championship rounds. In addition, 75% of the PIR valuation for small forwards cannot be explained by physical fitness alone. No predictive capacity of PIR is found for point guards during the two phases of the championship, as well as for forwards during the second round. These results indicate that for guards and forward players there are many more variables not related to PF that influence their performance in the game, while for point enters, PF can explain more than 50% of their performance. In the three statistically significant regressions, it is observed that the R2 value decreases in the second round, when the temporal distance from the realization of the physical fitness tests increases. The results of the analysis of variance show that the regression model results in a significantly better prediction of the PIR, when the predictors are used for each game position and moment of the season. The values of the Durbin–Watson test are within the normal parameters.

Table 8 shows the details of the model parameters and the importance of these values. In each model, the predictors are identified for each game position at each moment of the season. In addition, the results show that there is no collinearity between the model variables.

In all three statistically significant assumptions (first round forwards and first and second round center), the Player Load of the anaerobic test predicts a high PIR rating. The Player Load of the anaerobic test and the jump height in the ABK test for the forwards allow predicting the PIR only in the first round. These results highlight the importance of reactive work and the ability to withstand a greater load in anaerobic situations to achieve better performance in the game of this group of players.

On the other hand, in the predictive model of the center players we find the Player Load of the anaerobic capacity and the explosive distance in the centripetal force left both in the first and in the second lap. The ability to withstand a higher intensity load in anaerobic efforts and the ability to move explosively in curved movements help predict a better performance of center players.

Minutes of play were not included in the predictive model. There may be a relationship between a higher number of minutes played with a higher PIR. The more minutes played, the greater the possibility of performing positive actions that are included in the PIR calculation. However, the opposite may also occur. Moreover, there are cases in which female players play a few minutes but obtain a high PIR. The regression model calculated that includes minutes played as an independent variable increases the predictive power of FP on PIR, in all playing positions and during the two phases of the competition.

## 4. Discussion

Reviewing the existing literature on the study topics selected in this research, it was observed that there is a relationship between performance in competition and the PF of athletes [52], although it is unknown how these relationships evolve during a season. Therefore, the objectives of this research were to describe the PF level of the selected athletes according to the game position and to analyze its relationship with the GI of the competition (differentiating between the first and second round of the championship), finding significant differences in the PF level based on the game position and the relationship between PF and GI during the competition.

This research was conducted in an ecological context, in which the researchers did not alter or manipulate the participants’ interventions. The validity of the data is very relevant, since there are few investigations that provide information on the relationship between physical and game performance in female players of basketball. The power of the results of this type of research is significant since it analyzes an entire professional team [40].

### 4.1. Differences between PF According to Game Position

Carrying out a PF test or battery of tests allows knowing the physical-physiological state in which the athlete is. This generated knowledge allows the coaching staff to determine if the assimilation of loads, the training planning is correct or if, on the contrary, it requires modification [53]. In addition, it will provide a global vision of all the athletes and the profiles (strengths and limitations) of each athlete will be known, being able to use this in the planning of the following sessions [54]. Regarding the results obtained, no significant differences were observed in the tests carried out, except in the variable maximum speed (*T* Test with ball). In contrast to the findings, Mancha-Triguero, Reina, García-Rubio and Ibáñez [13] showed that guards and forwards of high-level women’s teams obtain better results than centers. For their part, Mancha-Triguero et al. [55] demonstrated during different assessments throughout the season in women’s basketball training that guard players are the ones who obtain better results than the rest of their teammates. These coincidences show the importance of guards and forwards presenting high levels of PF to obtain good results in the competition, as corroborated by Fernández-Cortes, Mandly, García-Rubio and Ibáñez [22] in the research on players in the Spanish women’s first division. If the tests carried out are analyzed, it can be observed that all players stand out in the tests related to strength and power (jumps and centripetal force). These results are opposite with those obtained by Hulka et al. [56] in a population of U18 male players, where they state that the game position directly influences the demands and requirements of the competition and therefore the intensity of the game or FP level. Gender, competitive level and age of the sample may determine differences in the physical fitness of basketball players according to game position.

### 4.2. Differences between GI According to Game Position

The results obtained show significant differences in most of the variables analyzed. The results of the Bonferroni Post Hoc analysis show that these differences generally occur between the guard and forward players with the center players. In this line, there are different documents [8,9,16,20] that coincide with the findings obtained in this investigation. This is mainly due to the fact that the physical characteristics of the players have a direct impact on the performances during the competition will be determined. The center players present higher values in the variables related to rebounds, scoring or efficacy in two-point shots [19,21,23], while the guard and forward players present better results in the variables related to the efficacy of two-point shots, three points, fouls received or assists among others [19,22,23]. These coincidences remain regardless of the competitive level or sex [20], since the organization of the sport facilitates hinders certain actions only because of their placement on the field of play (directly related to the game position) [21].

### 4.3. Relationship between GI and PF

The results obtained in the analysis of the bivariate correlations show that there is a relationship between the GI and the PF level of the analyzed team. The tests selected for the evaluation of the physical fitness show a direct relationship with the GI. These findings confirm that by analyzing the GI provided by the competition, the physical-physiological level of the athlete can be indirectly known. Coinciding with the findings of this research, Mancha-Triguero, Reina, García-Rubio and Ibáñez [13] showed that in high-level women’s basketball, Spanish first division, there is a relationship between PF and GI. To show this, they highlighted that aerobic capacity, anaerobic lactic acidity and centripetal force correlated with aspects of the competition, with aerobic capacity being the one with the greatest number of relationships. In line with these findings, Fernández-Cortes, Mandly, García-Rubio and Ibáñez [22] determined that two-point shots scored, stolen balls, fouls received and free throws scored are decisive in victory in the Spanish women’s first division. At each competitive level (first, second...division) [1], each type of competition (male or female) [57], playing context [58] or age of the players [59] shows specific performance indicators.

However, the results obtained on this occasion show that the multi-jump tests (power in the lower body), the anaerobic capacity and the agility test (*T* test) are the skills with the greatest influence on the competition. These differences with those existing in the literature may be due to the competitive level evaluated.

Regarding the detailed analysis of the relationships, the results show that the players with the best score stand out in aerobic capacity, multi-jumps, centripetal force and agility. Based on this profile, the sport of basketball itself is classified as a hybrid sport. This is because the player must have a high anaerobic capacity to deal with explosive actions and aerobic capacity to endure the duration of the match and help in the lactate clearance phase [60,61]. Another profile that we can find is that of the players with the greatest facility to score (two and three points shots and free throws). In this case, the players are characterized by lower values in the centripetal force test. In addition, scoring players are characterized by having a better level of aerobic capacity and anaerobic capacity. Chaouachi, et al. [62] determined that the best valued players were the ones that carried out the action’s explosives. These actions will be greater, of greater duration and of greater intensity. Finally, players with a PF profile in which defensive actions predominate, can be mentioned as such for their involvement in the game (they are divided into two groups: (i) defense close to the basket; (ii) defense away from the basket). In the case of defensive players close to the basket, they are characterized by contributing to the game in fouls committed, blocks and rebounds. In this line, these players generally show a negative proportional relationship between the mentioned GI and multi-jumps, repeated sprints and aerobic capacity. These technical–tactical actions are usually related to the game positions close to the basket. Due to this, these players must present high levels of strength and power that allow them to effectively carry out the defense actions that the competition demands of them [63]. On the other hand, the defensive players far from the basket stand out for their recoveries and defensive intensity, presenting a negative proportional relationship between agility and centripetal force. However, the players who made the greatest number of recoveries had high levels of aerobic and anaerobic lactic capacity. The PF profiles found in this research confirm the research by Gómez-Carmona, Mancha-Triguero, Pino-Ortega and Ibáñez [32] where they determined four PF profiles of basketball players. In the first PF profile, explosive efforts (jumps and centripetal force) stood out. This profile of explosive female players also appears in this research. The second PF profile stands out for finding the players with a high level of intermittent efforts. Regarding the similarity with this research, it can be observed that in the results obtained, there are scoring players who present high levels of anaerobic capacity. The third physiological profile, following the previous classification, refers to the players who stand out in actions where deceleration is an important factor. In this research, defensive players close to the basket stand out, since they have similar characteristics. Finally, the players with optimal results in continuous efforts or who stand out in the aerobic capacity test are the players who defend far from the basket. Figure 1 shows the PF and technical–tactical performance profiles, along with the positive relationship between these findings and those found by Gómez-Carmona, Mancha-Triguero, Pino-Ortega and Ibáñez [32] (Figure 3).

Finally, regarding the relationships between the PF level of the players analyzed and the GI of the competition, it was observed that there are differences between the matches belonging to the first and second round of the championship. Both Escudero-Tena, Rodríguez-Galan, García-Rubio and Ibáñez [21] and Fernández-Cortes, Mandly, García-Rubio and Ibáñez [22] showed that the GIs change between the regular and decisive phase in the women’s championship, where fewer errors are made, more throws and hits are made in the second than in the first. In this line, Gómez-Ruano, Lorenzo, Sampaio, Ibáñez and Ortega [23] showed that in close matches or those in which a clear objective is sought (the second phase of the championship is decisive because of the teams that play promotion play-offs and the relegation), they opt for different GI than when the match is not as important. This may be due to the fact that the level of PF has undergone some variation or that the dynamics of the matches have changed, since they already know each other from the first-round match. This knowledge of the rival team, added to the fact that it is the decisive phase of the championship, can cause the team to change its style of play or its characteristics.

### 4.4. Relationship between Physical and Game Performance Profiles by Specific Positions

Each playing position has a specific physical performance profile, which may be related to the roles coaches have assigned to them. The relationship between a higher score in the PIR of the guards’ female players with a better anaerobic capacity and a greater capacity to run distances in the aerobic test has been shown. The demands of the game require a high capacity to perform intermittent efforts, as well as to cover greater distances in these female players. Escudero-Tena, Rodríguez-Galan, García-Rubio and Ibáñez [21] identified the specific game actions that differentiate female players when winning and losing games. The one-, two-, and three-point shots, as well as assists were identified as key [15,22]. Therefore, coaches should plan specific high-intensity tasks that allow the female players to outperform their opponents, in order to achieve a higher rating.

The forward female players who achieve higher ratings do so when they obtain better values in jump height, travel more distance and act in an explosive way in the aerobic test. The ability to maintain continuous efforts at high intensity, moving from one side to the other in the front court to receive the ball, is very important for actions linked to a high PIR (shooting), as well as performing high intensity jumps, offensive and defensive rebounds. Total and defensive rebounds, as well as blocks made, free throws converted and assists are the actions that differentiate forwards when their teams win or lose [21,22]. These actions are linked to jumping height.

For their part, the center female players obtained better PIR scores when they obtained better results in the T-Tests with and without the ball, as well as covering greater explosive distances in the aerobic and anaerobic tests. These female players must make short cuts with continuous changes of direction in the spaces near the basket to gain position, which in turn will allow them to be effective in both the attack and defense phases. Their movements, in turn, must be at high intensity. The best center players when they win make more blocks, capture more defensive and total rebounds, score more two-pointers and assist their teammates [21,22]. These game indicators are linked to the ability to move quickly in tight spaces such as those used by these players.

Each type of female player requires specific physical training to improve the skills linked to the game actions that allow them to be more effective. Each coach must know which are the specific actions of effectiveness of the female players of his team, as these may vary depending on the style of play.

### 4.5. Predict the Relationship between PF and Player Performance Based on the Time of the Season

A low predictive capacity of the results of the physical fitness tests with the PIR has been identified depending on the playing position during the two phases of the season. The predictive capacity of the model has a greater impact on small forwards during the first phase of the season and on power forwards in both phases of the season. No impact of the CF on the PIR of point guards was identified at any time during the season. These findings confirm that there are other factors that condition the performance of players [64,65]. The individual technical and tactical mastery of the players, as well as the collective game, continue to be determinant for better individual performance [13,33]. In the analyzed sample, the center players have greater relevance of the physical aspect than technical–tactical aspects. In common to all playing positions, the predictive capacity of the model decreases as time increases, that is, it is lower in the second round of the championship.

The Player Load of the anaerobic test is the common physical capacity that contributes to predict a higher valuation of female forwards and center players. The ability to repeat high intensity efforts to overcome the intervention of their opponents is fundamental to achieve a better valuation [21,66]. In addition, small forwards improve their performance in the game if they have a high jumping ability. This fact was previously shown by Fernández-Cortes, Mandly, García-Rubio and Ibáñez [22], when he found that the small forward players who won the games differed from those who lost, in that they were able to capture more total and defensive rebounds, as well as receiving fewer blocks. All these game actions are related to jumping ability. Coaches can improve the performance of small forwards by working during the season on their ability to perform high-intensity actions repeatedly, as well as improving their jumping ability.

For center players, who usually play in spaces closer to the basket, the results of the centripetal strength test along with the Player Load anaerobic capacity test help predict PIR. The game actions that characterize inside players are explosive movements near the basket to obtain position, turns and changes of position to receive the ball [22,66,67]. These results show the relationship between the indicators of play by playing position and the physical abilities on which they are based. Coaches should adequately prepare their players, attending to a playing position so that they develop the best playing actions, individualizing the training [34].

Finally, the practical applications of this research were to facilitate physical work for the coach or coaching staff that can be individualized based on the playing position, confirming the importance of GI in PF. In addition, knowledge about specific PF is provided with the aim of improving certain individual profiles based on the characteristics of each player and her contribution to the game. GI analysis can be taken into account as a subjective measure of PF as long as the contextual characteristics (player, opposition level, team level, etc.) are similar.

The main limitation of the research is the analysis of a complete team. Although there are values that can be extrapolated to other teams, it is important that in order to take into account the results obtained, the sample is similar to that of this research. In addition, to carry out this research, data pertaining to a season is required.

Another limitation of the replication of this research is that to obtain part of the information, it is necessary to use high-cost technology, which is not always within the reach of all the teams. In spite of this, the battery of physical tests used allows the recording of different parameters without the need to use high-cost technological resources [29].

Finally, these results were obtained in a population of female basketball players, so their generalizability will be greater in this type of population.

## 5. Conclusions

The findings of this research corroborate the importance of PF in women’s basketball to achieve success. The results obtained show that there are different levels of PF in basketball players. These differences are strongly related to the game position of the game. Regarding the game position, significant differences are observed in the GIs obtained during the competition. This corroborates that the game position determines the contribution that a player makes to the team. In addition, the existence of four different physical-physiological profiles that are directly related to certain GIs is confirmed. Regarding the relationship between GI of the competition and PF, the results show positive relationships between both variables, being able to affirm that the achievement of certain GI in competition is influenced by the level of PF in some specific skill or capacity. Regarding the prediction analysis carried out, it is concluded that the level of PF determines the performance in competition differently depending on the position, namely, higher on guards and centers and lower on forwards and depending on the moment of the season (the first round having greater impact).

## Figures and Tables

**Figure 1 ijerph-20-00988-f001:**
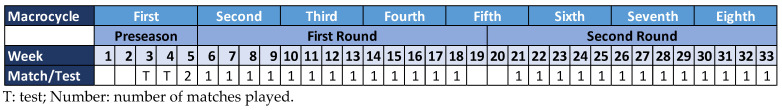
Data collection phases.

**Figure 2 ijerph-20-00988-f002:**
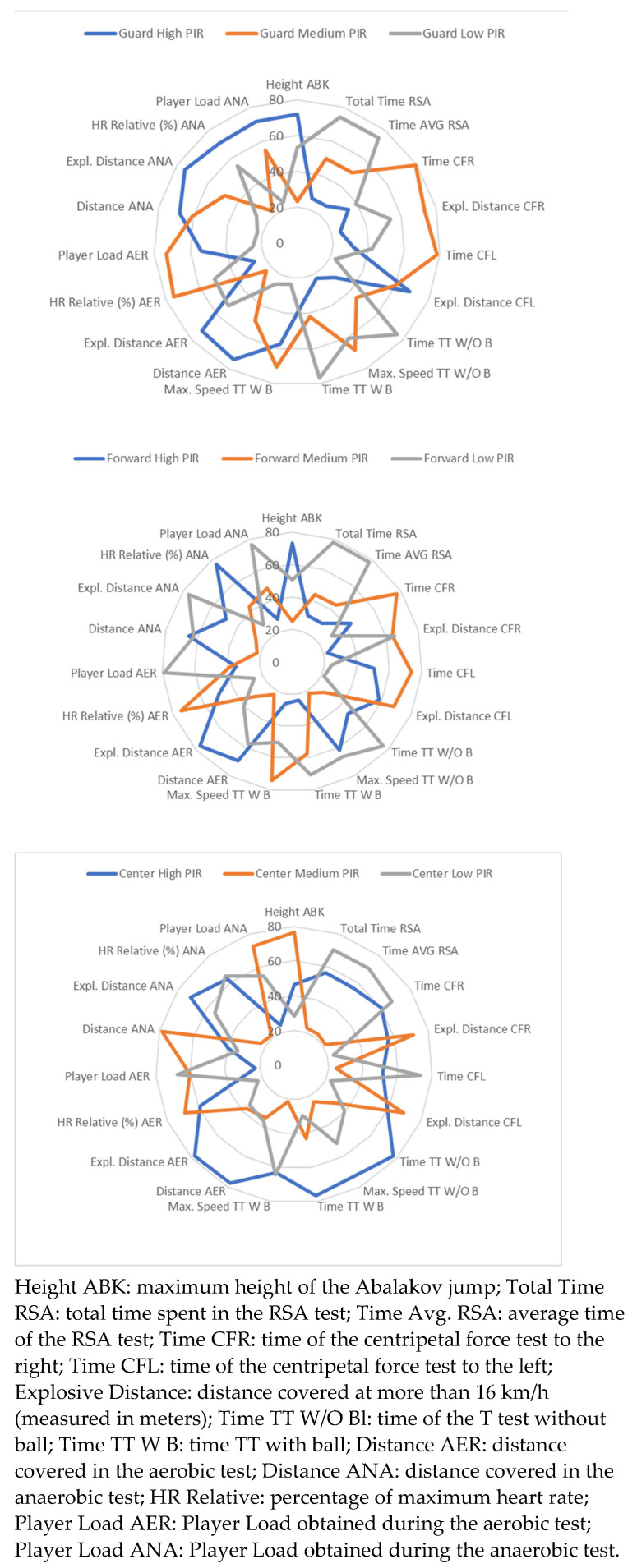
Physical profiles of players according to their performance in the game, by playing positions.

**Figure 3 ijerph-20-00988-f003:**
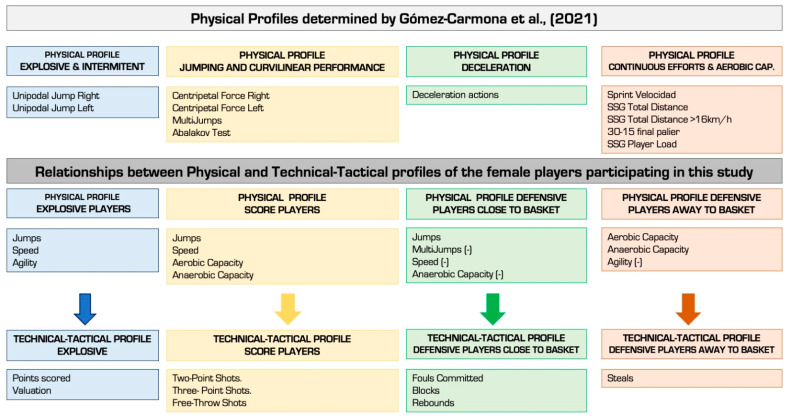
Representation of the relationships between physical and technical–tactical profiles [32].

**Table 1 ijerph-20-00988-t001:** Normality test results.

Physical Fitness Variables		*S–W*	*df*	*p*	Game Indicators Variables	*K–S ^a^*	*df*	*p*
Abalakov	Height	0.879	9	0.154	Points/min	0.086	228	0.000 *
Multi-jump	Height	0.916	9	0.358	Efficacy 2 points/min	0.345	228	0.000 *
	Between Jump	0.936	9	0.535	Efficacy 3 points/min	0.342	228	0.000 *
RSA Test	Total Time	0.968	9	0.875	Efficacy Free Throw/min	0.304	228	0.000 *
	Avg. Time	0.968	9	0.881	Committed Fouls/min	0.117	228	0.000 *
Centripetal force right	Time	0.881	9	0.160	Received Fouls/min	0.126	228	0.000 *
	Expl. Distance	0.845	9	0.065	Assists/min	0.158	228	0.000 *
Centripetal force left	Time	0.917	9	0.367	Blocks/min	0.454	228	0.000 *
	Expl. Distance	0.855	9	0.085	Defensive Rebounds/min	0.131	228	0.000 *
*T* Test Without Ball	Time	0.974	9	0.926	Offensive Rebounds/min	0.229	228	0.000 *
	Max. Speed	0.660	9	0.000 *	Total Rebounds/min	0.136	228	0.000 *
*T* Test with Ball	Time	0.936	9	0.539	Steals/min	0.257	228	0.000 *
	Max. Speed	0.952	9	0.715	Turnovers/min	0.134	228	0.000 *
Aerobic capacity	Distance	0.971	9	0.901	PIR/min	0.069	228	0.010 *
	Expl. Distance	0.839	9	0.057				
	HR Relative (%)	0.799	9	0.020 *				
	Player Load	0.962	9	0.823				
Anaerobic capacity	Distance	0.858	9	0.091				
	Expl. Distance	0.973	9	0.917				
	HR Relative (%)	0.912	9	0.331				
	Player Load	0.954	9	0.734				

*^a^*: Lilliefors significance correction; *S–W*: Shapiro–Wilk test; *K–S*: Kolmogorov–Smirnov test; *df*: degree of freedom; * significant variable (*p* < 0.05).

**Table 2 ijerph-20-00988-t002:** Sociodemographic and body composition characteristics of the sample.

	Teams		Guards (*n* = 3)		Forwards (*n* = 5)		Centers (*n* = 4)	
	*Mean*	*SD*	*Mean*	*SD*	*Mean*	*SD*	*Mean*	*SD*
Age	25.25	±7.617	20.00	±1.000	29.00	±11.832	25.40	±4.393
Height	178.25	±9.206	169.67	±3.215	174.25	±3.403	186.60	±7.797
Body mass	72.33	±11.657	60.33	±4.509	67.50	±2.646	83.40	±8.591

**Table 3 ijerph-20-00988-t003:** Descriptive results of physical fitness tests grouped by game position and analysis of the differences between female players.

		Guards (*n* = 3)		Forwards (*n* = 5)		Centers (*n* = 4)				*Post Hoc*	η^2^
		*Mean*	*SD*	*Mean*	*SD*	*Mean*	*SD*	*F*	*Sig.*		
Abalakov	Height (cm)	33.34	±15.59	34.43	±14.65	34.43	±14.65	0.052	0.950		0.021
Multi-jump	Height (cm)	28.17	±4.29	31.79	±4.93	30.03	±6.66	0.338	0.726		0.114
	Between Jump (ms)	450.25	±51.97	486.08	±47.66	435.08	±55.91	0.762	0.507		0.180
RSA Test	Total Time (s)	12.66	±0.22	13.20	±0.25	12.98	±0.70	1.118	0.387		0.244
	Avg. Time (s)	2.53	±0.04	2.64	±0.05	2.60	±0.14	1.115	0.338		0.243
F Centríp. Right	Time (s)	4.70	±0.07	4.68	±0.05	4.84	±0.08	5.159	0.050		0.627
	Expl. Distance (m)	4.08	±3.85	4.76	±4.02	3.25	±2.95	0.129	0.881		0.033
F. Centríp. Left	Time (s)	4.59	±0.11	4.66	±0.04	4.82	±0.21	1.979	0.219		0.409
	Expl. Distance (m)	5.81	±1.29	4.75	±0.27	3.71	±2.40	1.328	0.333		0.415
*T* Test without ball	Time (s)	12.25	±0.54	12.56	±0.38	12.51	±0.21	0.535	0.611		0.130
	Max. Speed (km/h)	18.03	±0.90	17.73	±1.89	16.72	±1.58	0.359	0.713		0.110
*T* Test with ball	Time (s)	12.96	±0.87	12.91	±0.25	13.57	±0.78	0.855	0.471		0.223
	Max. Speed (km/h)	18.50	±0.39	20.01	±1.21	17.35	±0.62	8.064	0.020 *	** *β* **	0.689
Aerobic capacity	Distance (m)	1949.69	±77.33	1830.44	±118.08	1843.18	±92.88	1.352	0.328		0.294
	Expl. Distance (m)	73.93	±20.76	59.02	±18.63	65.12	±27.53	0.329	0.732		0.093
	HR Relative (%) (bpm)	94.60	±0.10	93.33	±1.35	92.50	±2.21	1.497	0.297		0.323
	Player Load (a.u)	26.94	±2.67	24.97	±2.78	25.81	±0.55	0.580	0.589		0.157
Anaerobic capacity	Distance (m)	810.66	±18.09	793.45	±11.57	753.99	±47.12	2.833	0.136		0.480
	Expl. Distance (m)	159.16	±17.88	136.89	±14.40	125.44	±11.71	3.985	0.079		0.569
	HR Relative (%) (bpm)	73.97	±4.74	82.03	±0.51	80.67	1±0.89	1.187	0.368		0.261
	Player Load (a.u.)	12.61	±1.42	12.37	±1.32	11.66	±1.31	0.402	0.686		0.110

Height: maximum jump height; Total Time: total time spent in the 5-sprint carried out; Expl. Distance: explosive distance (>18 km/h); Max. Speed: maximum speed reached; HR Relative: % of heart rate maximum; α: significant differences between guards and forwards; β: significant differences between forwards and centers; δ: Significant differences between Guards and Centers; cm: centimeters; ms: milliseconds; s: seconds; m: meters; km/h: kilometers per hour; bpm: beats per minute; a.u.: arbitrary unit. * significant variable (*p* < 0.05).

**Table 4 ijerph-20-00988-t004:** Descriptive and inferential results of game indicators relativized per minute grouped by game position and the differences between female players.

	Guards (*n* = 3)		Forwards (*n* = 5)		Centers (*n* = 4)				
	*Mean*	*SD*	*Mean*	*SD*	*Mean*	*SD*	*H*	*Sig*	*Post Hoc*
Points/min	0.30	±0.18	0.27	±0.20	0.32	±0.21	3.234	0.198	
Efficacy 2 points/min	0.02	±0.03	0.02	±0.03	0.03	±0.09	3.970	0.137	
Efficacy 3 points/min	0.02	±0.03	0.04	±0.08	0.01	±0.02	33.987	<0.001 *	** *β δ* **
Efficacy Free Throw/min	0.04	±0.06	0.02	±0.04	0.05	±0.10	12.018	0.002 *	** *β* **
Committed Fouls/min	0.09	±0.06	0.11	±0.10	0.13	±0.10	4.661	0.097	
Received Fouls/min	0.12	±0.09	0.06	±0.06	0.10	±0.08	22.120	<0.001 *	** *α β* **
Assists/min	0.07	±0.06	0.06	±0.06	0.06	±0.06	2.826	0.243	
Blocks/min	0.01	±0.02	0.01	±0.03	0.02	±0.04	13.863	0.001 *	** *β δ* **
Defensive Rebounds/min	0.09	±0.07	0.05	±0.06	0.23	±0.11	94.271	<0.001 *	** *α β δ* **
Offensive Rebounds/min	0.03	±0.04	0.04	±0.04	0.10	±0.11	31.707	<0.000 *	** *β δ* **
Total Rebounds/min	0.12	±0.09	0.09	±0.09	0.33	±0.18	89.532	<0.001 *	** *β δ* **
Steals/min	0.07	±0.06	0.03	±0.04	0.05	±0.07	11.281	0.004 *	** *α* **
Turnovers/min	0.11	±0.07	0.07	±0.07	0.07	±0.06	14.103	0.001 *	** *α δ* **
PIR/min	0.23	±0.25	0.10	±0.32	0.47	±0.29	45.706	<0.001 *	** *α β δ* **

Efficacy 2 Points = 2-P shots annotated/2-P shots tried; Efficacy 3 Points = 3-P shots annotated/3-P shots tried; Efficacy Free Throw = free throw shots annotated/free throw shots tried; α: significant differences between guards and forwards; β: significant differences between forwards and centers; δ: significant differences between guards and centers; *: significant variable (*p* < 0.05).

**Table 5 ijerph-20-00988-t005:** (**a**) Results of the bivariate correlations between physical fitness and game indicators of the competition relativized per minute in the first round of the season. (**b**) Results of the bivariate correlations between physical fitness and game indicators of the competition relativized per minute in the second round of the season.

**(a)**																	
First Round				**Points**	**Efficacy 2-P Shots**	**Efficacy 3-P Shots**	**Efficacy Free Throw Shots**	**Committed Fouls**	**Received Fouls**	**Assists**	**Blocks**	**Defensives Rebounds**	**Offensives Rebounds**	**Total Rebounds**	**Steals**	**Turnovers**	**PIR**
	MJ	Height	rho	0.068	0.030	−0.048	−0.038	**0.402**	−0.105	**−0.238**	0.115	−0.073	**0.198**	0.002	**−0.303**	−0.003	−0.178
			*p*	0.474	0.750	0.616	0.687	**0.000**	0.269	**0.011**	0.224	0.444	**0.036**	0.979	**0.001**	0.977	0.060
		Between Jump	rho	−0.177	−0.173	0.176	−0.109	**−0.260**	**−0.290**	0.048	**−0.253**	**−0.401**	**−0.343**	**−0.433**	0.059	−0.139	**−0.239**
			*p*	0.060	0.067	0.062	0.251	**0.005**	**0.002**	0.614	**0.007**	**0.000**	**0.000**	**0.000**	0.537	0.141	**0.011**
	RSA	Total Time	rho	**−0.233**	−0.002	−0.116	−0.113	−0.161	**−0.305 ***	0.094	−0.107	−0.030	−0.131	−0.055	0.098	**−0.252**	−0.108
			*p*	**0.013**	0.984	0.220	0.233	0.089	**0.001**	0.321	0.257	0.755	0.166	0.565	0.301	**0.007**	0.257
	CFR	Time	rho	0.033	0.025	**−0.330**	0.052	0.167	0.085	−0.072	**0.222**	**0.518**	0.085	**0.451**	0.032	0.014	**0.214**
			*p*	0.729	0.795	**0.000**	0.581	0.078	0.369	0.446	**0.018**	**0.000**	0.369	**0.000**	0.734	0.884	**0.023**
		Expl. Distance	rho	0.170	0.030	**0.193**	0.007	0.119	**0.245**	**−0.253**	0.053	−0.174	0.063	−0.115	**−0.236**	0.184	−0.137
			*p*	0.072	0.755	**0.041**	0.945	0.211	**0.009**	**0.007**	0.577	0.066	0.505	0.226	**0.012**	0.052	0.147
	CFL	Time	rho	−0.124	0.043	**−0.247**	−0.083	0.143	−0.092	−0.018	0.091	**0.268**	−0.048	**0.214**	0.093	−0.166	0.016
			*p*	0.192	0.651	**0.008**	0.379	0.132	0.332	0.854	0.336	**0.004**	0.615	**0.023**	0.330	0.079	0.864
		Expl. Distance	rho	0.120	0.037	0.147	0.007	**0.312**	**0.221**	−0.112	0.030	−0.119	−0.042	−0.113	0.008	0.066	−0.008
			*p*	0.207	0.695	0.121	0.942	**0.001**	**0.018**	0.238	0.754	0.208	0.657	0.232	0.937	0.489	0.931
	TT without Ball	Time	rho	−0.089	0.089	−0.086	−0.007	−0.170	−0.179	0.039	−0.048	−0.018	0.150	0.039	−0.102	−0.073	−0.083
			*p*	0.346	0.349	0.366	0.941	0.072	0.058	0.682	0.613	0.853	0.112	0.679	0.282	0.443	0.384
	TT with Ball	Time	rho	0.060	0.091	**−0.187**	0.087	0.094	−0.006	**−0.192**	0.096	0.180	**0.205**	**0.216**	**−0.222**	0.052	0.006
			*p*	0.525	0.338	**0.047**	0.357	0.321	0.948	**0.042**	0.310	0.056	**0.029**	**0.022**	**0.018**	0.583	0.951
		Max. Speed	rho	−0.114	−0.142	**0.379**	**−0.188**	−0.058	−0.109	−0.019	**−0.233**	**−0.593**	**−0.357**	**−0.591**	0.008	−0.028	**−0.410**
			*p*	0.229	0.134	**0.000**	**0.047**	0.541	0.249	0.842	**0.013**	**0.000**	**0.000**	**0.000**	0.933	0.769	**0.000**
	AER	Distance	rho	**−0.199**	0.147	0.025	−0.072	0.109	0.029	**0.213**	−0.168	−0.074	**−0.227**	−0.129	**0.347**	−0.100	−0.097
			*p*	**0.034**	0.120	0.790	0.447	0.249	0.759	**0.023**	0.075	0.433	**0.016**	0.173	**0.000**	0.293	0.309
		Expl. Distance	rho	**−0.197**	0.174	−0.037	−0.061	0.176	−0.062	0.159	−0.147	−0.072	−0.123	−0.094	**0.239**	−0.127	−0.130
			*p*	**0.036**	0.065	0.696	0.520	0.062	0.515	0.092	0.121	0.450	0.196	0.320	**0.011**	0.180	0.170
		HR Relative (%)	rho	0.080	−0.156	**0.278**	−0.043	**0.203**	0.153	−0.184	−0.059	**−0.306**	**−0.256**	**−0.335**	−0.092	**0.246**	**−0.280**
			*p*	0.399	0.100	**0.003**	0.654	**0.031**	0.106	0.051	0.537	**0.001**	**0.006**	**0.000**	0.330	**0.009**	**0.003**
		Player Load	rho	0.118	0.121	0.039	0.108	**−0.201**	**0.377**	0.094	0.046	**0.198**	0.053	0.182	0.177	0.086	**0.305**
			*p*	0.214	0.201	0.685	0.254	**0.033**	**0.000**	0.323	0.625	**0.036**	0.580	0.054	0.061	0.364	**0.001**
	ANA	Distance	rho	0.163	0.057	**0.214**	0.056	0.053	**0.300**	0.017	0.017	−0.111	0.070	−0.074	0.019	0.152	0.082
			*p*	0.085	0.552	**0.023**	0.557	0.579	**0.001**	0.855	0.861	0.243	0.459	0.438	0.841	0.108	0.389
		Expl. Distance	rho	−0.136	0.035	**0.271**	−0.073	−0.183	0.108	**0.239**	**−0.246**	**−0.303**	**−0.297**	**−0.339**	**0.301**	0.025	−0.145
			*p*	0.152	0.711	**0.004**	0.443	0.053	0.257	**0.011**	**0.008**	**0.001**	**0.001**	**0.000**	**0.001**	0.795	0.125
		HR Relative (%)	rho	**−0.311**	−0.003	**−0.186**	−0.152	0.002	**−0.415**	0.141	−0.122	−0.012	**−0.194**	−0.063	**0.186**	**−0.348**	−0.131
			*p*	**0.001**	0.979	**0.048**	0.107	0.981	**0.000**	0.135	0.200	0.903	**0.040**	0.509	**0.048**	**0.000**	0.167
		Player Load	rho	0.162	0.001	**0.225**	0.067	−0.156	**0.264**	0.019	−0.010	−0.120	0.014	−0.095	0.040	0.077	0.170
			*p*	0.086	0.988	**0.016**	0.483	0.099	**0.005**	0.844	0.915	0.205	0.879	0.315	0.673	0.417	0.072
**(b)**																	
Second Round				**Points**	**Efficacy 2-P Shots**	**Efficacy 3-P Shots**	**Efficacy Free Throw Shots**	**Committed Fouls**	**Received Fouls**	**Assists**	**Blocks**	**Defensives Rebounds**	**Offensives Rebounds**	**Total Rebounds**	**Steals**	**Turnovers**	**PIR**
	MJ	Height	rho	0.046	0.085	−0.002	−0.047	**0.328**	−0.124	**−0.218**	0.070	−0.100	**0.209**	−0.012	**−0.240**	−0.015	−0.147
			*p*	0.625	0.366	0.980	0.620	**0.000**	0.187	**0.019**	0.459	0.286	**0.025**	0.902	**0.010**	0.874	0.116
		Between Jump	rho	−0.178	−0.123	0.126	−0.153	**−0.222**	**−0.280**	0.077	**−0.214**	**−0.375**	**−0.353**	**−0.418**	−0.047	−0.128	**−0.261**
			*p*	0.057	0.190	0.178	0.102	**0.017**	**0.002**	0.414	**0.022**	**0.000**	**0.000**	**0.000**	0.615	0.172	**0.005**
	RSA	Total Time	rho	**−0.221**	−0.054	−0.025	−0.002	−0.097	**−0.287**	0.106	−0.065	−0.010	−0.150	−0.051	−0.010	**−0.227**	−0.149
			*p*	**0.018**	0.569	0.794	0.983	0.301	**0.002**	0.260	0.489	0.916	0.109	0.590	0.915	**0.015**	0.111
	CFR	Time	rho	0.070	0.060	−0.181	0.115	0.171	0.099	−0.031	**0.222**	**0.526**	0.137	**0.471**	0.061	0.013	**0.284**
			*p*	0.457	0.522	0.053	0.221	0.067	0.295	0.743	**0.017**	**0.000**	0.146	**0.000**	0.514	0.889	**0.002**
		Expl. Distance	rho	0.151	0.054	0.100	**−0.194**	0.089	**0.223**	**−0.315**	0.059	**−0.184**	0.019	−0.135	**−0.196**	0.146	−0.159
			*p*	0.108	0.569	0.285	**0.038**	0.345	**0.017**	**0.001**	0.529	**0.049**	0.842	0.151	**0.036**	0.120	0.090
	CFL	Time	rho	−0.080	0.034	−0.056	−0.028	0.176	−0.069	−0.002	0.132	**0.297**	−0.017	**0.242**	0.059	−0.158	0.056
			*p*	0.397	0.717	0.554	0.767	0.060	0.462	0.980	0.159	**0.001**	0.856	**0.009**	0.531	0.091	0.552
		Expl. Distance	rho	0.119	0.101	0.130	**−0.206**	**0.250**	**0.211**	−0.126	0.014	−0.130	−0.033	−0.114	0.090	0.044	0.032
			*p*	0.207	0.282	0.166	**0.027**	**0.007**	**0.024**	0.180	0.884	0.167	0.724	0.224	0.339	0.638	0.731
	TT without Ball	Time	rho	−0.126	−0.003	−0.132	**0.184**	−0.140	**−0.189**	0.021	−0.053	−0.025	0.096	0.015	−0.150	−0.063	−0.167
			*p*	0.181	0.972	0.161	**0.049**	0.136	**0.043**	0.826	0.575	0.791	0.309	0.870	0.109	0.504	0.074
	TT with Ball	Time	rho	0.028	0.115	**−0.247**	0.173	0.069	−0.025	**−0.184**	0.061	0.155	**0.183**	**0.190**	**−0.209**	0.048	−0.020
			*p*	0.765	0.221	**0.008**	0.064	0.465	0.792	**0.049**	0.515	0.099	**0.050**	**0.042**	**0.025**	0.613	0.835
		Max. Speed	rho	−0.097	−0.105	**0.342**	**−0.339**	−0.045	−0.097	−0.056	−0.169	**−0.558**	**−0.373**	**−0.569**	−0.048	−0.038	**−0.412**
			*p*	0.301	0.265	**0.000**	**0.000**	0.630	0.302	0.550	0.071	**0.000**	**0.000**	**0.000**	0.614	0.684	**0.000**
	AER	Distance	rho	−0.178	0.112	0.033	0.061	0.137	0.050	0.176	−0.131	−0.046	**−0.228**	−0.105	**0.360**	−0.087	−0.099
			*p*	0.057	0.233	0.724	0.516	0.143	0.597	0.060	0.164	0.628	**0.014**	0.266	**0.000**	0.357	0.295
		Expl. Distance	rho	**−0.197**	0.142	−0.034	0.126	**0.183**	−0.053	0.138	−0.138	−0.067	−0.132	−0.090	**0.259**	−0.109	−0.150
			*p*	**0.035**	0.131	0.719	0.180	**0.050**	0.576	0.141	0.141	0.479	0.159	0.336	**0.005**	0.245	0.110
		HR Relative (%)	rho	0.107	0.001	**0.200**	**−0.187**	0.150	0.157	**−0.189**	−0.040	**−0.293**	**−0.216**	**−0.309**	−0.036	**0.222**	**−0.197**
			*p*	0.254	0.996	**0.032**	**0.046**	0.109	0.094	**0.044**	0.674	**0.001**	**0.021**	**0.001**	0.700	**0.017**	**0.035**
		Player Load	rho	0.114	0.030	−0.017	−0.022	−0.157	**0.369**	0.025	0.064	**0.193**	0.004	0.162	0.183	0.074	**0.245**
			*p*	0.224	0.751	0.854	0.812	0.095	**0.000**	0.787	0.495	**0.038**	0.964	0.083	0.051	0.432	**0.008**
	ANA	Distance	rho	0.144	0.031	0.129	−0.095	0.025	**0.285**	−0.031	−0.002	−0.122	0.041	−0.087	0.101	0.129	0.071
			*p*	0.124	0.740	0.170	0.314	0.794	**0.002**	0.746	0.986	0.192	0.661	0.353	0.283	0.168	0.448
		Expl. Distance	rho	−0.117	−0.022	0.181	−0.044	−0.132	0.126	0.168	**−0.186**	**−0.267**	**−0.326**	**−0.318**	**0.281**	0.030	−0.178
			*p*	0.213	0.814	0.052	0.640	0.159	0.181	0.073	**0.047**	**0.004**	**0.000**	**0.001**	**0.002**	0.752	0.056
		HR Relative (%)	rho	**−0.283**	−0.023	−0.023	0.011	0.043	**−0.383**	**0.187**	−0.088	0.010	−0.178	−0.046	0.094	**−0.309**	−0.127
			*p*	**0.002**	0.805	0.810	0.908	0.648	**0.000**	**0.046**	0.348	0.916	0.057	0.626	0.320	**0.001**	0.175
		Player Load	rho	0.138	−0.032	0.133	**−0.194**	−0.157	**0.246**	−0.023	−0.017	−0.136	−0.031	−0.121	0.055	0.056	0.124
			*p*	0.141	0.737	0.157	**0.038**	0.094	**0.008**	0.804	0.853	0.148	0.741	0.199	0.559	0.549	0.188

MJ: Multi-jump test; RSA: Repeat Sprint Ability; CFR: centripetal force to the right; CFL: centripetal force to the left; T T w/out: *T* Test without ball; T T with ball: *T* Test with ball; AER: aerobic capacity; ANA: anaerobic capacity; PIR: Performance Index Rating. * *p* > 0.05; rho: Spearman’s correlation coefficient; Colour green: the relationships in which they coincide in the first and the second round; Colour orange: the relationships only occur in one round.

**Table 6 ijerph-20-00988-t006:** Physical and game performance profiles by playing positions.

	Guards			Forwards			Centers		
	*PIR*	*%*	*p*	*PIR*	*%*	*p*	*PIR*	*%*	*p*
High PIR	7.46	34.2	<0.001	4.08	32.0	<0.001	16.68	32.5	<0.001
Medium PIR	6.04	31.6		3.38	34.7		14.73	33.8	
Low PIR	3.69	34.2		−0.32	33.3		3.38	33.8	

PIR: Performance Index Rating.

**Table 7 ijerph-20-00988-t007:** Results of the prediction model.

		*R*	*R* ^2^	*SE*	*Durbin–Watson*	*F*	*p*
Guard	First round	0.266 ^a^	0.071	5.997	1.417	1.296	0.287 ^a^
	Second round	0.264 ^a^	0.070	5.934	1.398	1.352	0.271 ^a^
Forward	First round	0.496 ^b^	0.246	4.175	1.283	5.705	0.007 ^b,^*
	Second round	0.373 ^b^	0.139	4.195	1.349	2.745	0.079 ^b^
Center	First round	0.733 ^c^	0.538	5.843	1.848	20.350	0.000 ^c,^*
	Second round	0.697 ^c^	0.486	6.126	1.593	17.035	0.000 ^c,^*

^a^ Predictors: (Constant), Player Load (anaerobic capacity), Between Jump (MJ); ^b^ Predictors: (Constant), Player Load (anaerobic capacity), Height ABK; ^c^ Predictors: (Constant), Player Load (anaerobic capacity), explosive distance (centripetal force left). * *p* > 0.05.

**Table 8 ijerph-20-00988-t008:** Parameter analysis.

Position			Unstandardized Coefficients		Standardized Coefficients	*t*	*p*	95.0% Confidence Interval for B	
			*B*	*SE B*	β			Lower Bound	Upper Bound
Guards	First round	(Constant)	−12.194	16.231		−0.751	0.458	−45.179	20.790
		Between Jump (MJ)	0.002	0.024	0.015	0.091	0.928	−0.047	0.051
		Player Load (Ana Cap.)	1.340	0.835	0.268	1.604	0.118	−0.357	3.038
	Second round	(Constant)	−10.909	15.631		−0.698	0.490	−42.611	20.792
		Between Jump (MJ)	−0.001	0.023	−0.004	−0.023	0.982	−0.046	0.045
		Player Load (Ana Cap.)	1.342	0.826	0.264	1.623	0.113	−0.334	3.018
Forwards	First round	(Constant)	−24.150	9.590		−2.518	0.017	−43.618	−4.681
		Player Load (Ana Cap.)	1.632	0.693	0.376	2.355	0.024	0.225	3.038
		Height ABK	0.192	0.061	0.504	3.155	0.003	0.068	0.316
	Second round	(Constant)	−11.523	9.757		−1.181	0.246	−31.350	8.305
		Player Load (Ana Cap.)	0.694	0.700	0.173	0.992	0.328	−0.728	2.117
		Height ABK	0.147	0.063	0.408	2.343	0.025	0.019	0.275
Centers	First round	(Constant)	−51.938	10.479		−4.957	0.000	−73.211	−30.665
		Player Load (Ana Cap.)	5.647	0.893	0.728	6.324	0.000	3.834	7.460
		Expl. Dist. Cent. Left	−0.576	0.484	−0.137	−1.190	0.242	−1.559	0.407
	Second round	(Constant)	−46.171	10.806		−4.273	0.000	−68.087	−24.256
		Player Load (Ana Cap.)	5.235	0.922	0.680	5.676	0.000	3.364	7.106
		Expl. Dist. Cent. Left	−0.914	0.502	−0.218	−1.820	0.077	−1.933	0.105

MJ: multi-jump test; Ana Cap: anaerobic capacity; Expl. Dist. Cent. Left: explosive distance centripetal test left.

## Data Availability

Not applicable.

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
