# Peer review of "Physical Fitness as a Predictor of Performance during Competition in Professional Women’s Basketball Players"

_ijerph, 2023, doi:10.3390/ijerph20020988_

Round 1
Reviewer 1 Report (New Reviewer)
Please follow the comments under each of the headings. This will allow the study to have greater credibility and scientific rigour. The added figure under discussion is very complete but does not include the abalakov variable.

Author Response
Reviewer 1
Dear reviewer, the authors appreciate your work and the time spent in reviewing the manuscript. Below we detail and modify all the questions you have requested. The authors hope that they are to your liking.
Their contributions have improved the manuscript qualitatively
To facilitate their work, the authors have marked in red all the modifications made to the manuscript.
STRUCTURE
- In general, the manuscript is properly structured
The authors appreciate your feedback
TITLE AND ABSTRACT
- The title or abstract should give information on what kind of study it is about. It is also advisable to add the population where the study has been carried out.
We appreciate your suggestion. The authors consider that the title includes the requested information. However, to make it even more specific, the term "professional" has been included in the title.
- Each section can be highlighted in bold, for example Background: “The evaluation of physical fitness in team sports is enjoying greater importance in the training of professional…”. In the same vein, materials and method, results and conclusions would be indicated.
We find the inclusion of sections within the summary very interesting. The authors have not done so, as the journal's guidelines recommend the use of "a single paragraph of no more than 200 words". In the case of research articles, abstracts should provide a relevant overview of the work. We strongly encourage authors to use the following style of structured abstracts, but without titles."
INTRODUCTION
- Very good start to the introduction, a good description of the sport and different areas are covered.
The authors appreciate your feedback
- The introduction is very short and an attempt should be made to update quotations. The hypothesis of the study should be added at the end of the introduction.
Thank you very much for your comment that substantially improves the quality of the manuscript. To this end, the Introduction has been improved and the references have been updated. In addition, hypotheses have been added to the objectives of this research.
New paragraphs have been included in the introduction section.
“Research approaches are sometimes conducted in isolation, without interaction be-tween the different topics. The evolution of basketball research requires interdiscipli-nary studies in which more than one object of study is related, as this sport is complex, in which multiple variables interact. This research aims to identify the relationship be-tween two of them, analyzing the relationship that may exist between physical condi-tion and game indicators.”
“To recruit players for a team, coaches use various objective and subjective infor-mation. Among the objective information, they use the Performance Index Rating (PIR), which is provided by the official competitions and is public [37]. This information can be classified according to the specific position. There is little public information on the physical performance indicators of the players that would allow an adequate selec-tion of players according to the needs of the game. Only in the NBA are rookie players given physical and technical tests, the Draft Combine. The results of these tests are not always directly related to subsequent performance during their first year of competi-tion [38].”
- It is advisable to add the hypothesis of the study at the end of the introduction.
The following paragraph has been included at the end of the introduction section
“Based on these objectives, the following hypotheses are made: a) there will be differences in physical fitness results among female players according to their specific position; b) there will be differences in game indicators among female players of different specific positions; c) during each moment of the season there will be specific relationships between physical fitness and game indicators; d) there will be some physical fitness test results that can predict the performance of female players.”
- Line 35: “These investigations have different objects … physical fitness (PF) [10-11] or game indicators (GI) [12-14]”. It might be useful to clarify the following paragraph.
Its appreciation is interesting, as the paragraph is very brief. It has been complemented by including the following phases:
“Research approaches are sometimes conducted in isolation, without interaction between the different topics. The evolution of basketball research requires interdisciplinary studies in which more than one object of study is related, as this sport is complex, in which multiple variables interact. This research aims to identify the relationship between two of them, analyzing the relationship that may exist between physical fitness and game indicators.”
- The literature search is brief. In addition, some references are more than five years old. Please try to update the quotations.
The authors appreciate your comment. New references have been added and those that were more than 5 years old have been updated. The authors hope you like
Other references older than 5 years have been retained due to their relevance.
MATERIAL AND METHODS
2.1. Design
- It does not describe the design. Specify the type of study. For example: "A descriptive, cross-sectional design was used to analyse".
We appreciate your suggestion. The following sentence has been included at the end of the design paragraph:
“Specifically, this is an empirical study, with a quantitative, descriptive and cross-sectional methodology.”
2.2. Participants
- The results of the sample, such as age, weight, height, etc., should not be included in the methodology section. These values go in the results section. Here it is advisable to specify how the participants were collected.
Your comment is very interesting. The authors have kept this basic information in this section, as it is usual in similar works.
The participants section has been substantially modified following their recommendations.
The following paragraphs have been included
“Convenience sampling was used, as it is a non-probabilistic and non-random sampling technique. This sample was selected according to the ease of access offered by the club and the availability of the female players to be part of the sample. The re-searchers selected this sample because the team was already participating in another research on the quantification of training and competition loads. The head coach was contacted first to find out his availability. After agreeing to participate in the study, permission was sought from the club.
A professional women's basketball team, which participates in the Spanish Liga Femenina 2, participated in this study. The team was composed of 13 female players. The reality of sports training is that the number of players analyzed is always relatively low, since it is a high competition context. This reality of the high-level sports context does not limit the validity of its results [41]. Finally, only 12 female players were included in the study as they met the inclusion criteria.”
“2.2.1. Eligibility criteria.
The following criteria were used to select the cases participating in the sample: i) Only female basketball players belonging to the club analyzed were included in this study; ii) To be considered a case study, the female player must have participated in at least 12 minutes per game [42]; iii) having played at least 1year professional category; iv) having performed the physical fitness tests during the pre-season; v) having completed the two weeks of previous training, vi) not having suffered injuries in the first weeks of training.
Exclusion criteria were: i) participating less than 12 minutes of play in each match analyzed; ii) not having actively participated in the physical condition test, iii) being involved in another type of sport at the same time as in this study; iv) not having a sufficient cognitive level to collaborate in the study or not being able to attend the established sessions.”
Nevertheless, we have included at the beginning of the results section a table with the sociodemographic and anthropometric characteristics of the female players.
- Describe the setting, locations, and relevant dates, including periods of recruitment, exposure, follow-up, and data collection information. All persons who have participated should be reported in the study, not only the participants.
We appreciate your comments. The Sample section has been improved with the following paragraph:
“The data were collected during the 2020-2021 season. Physical tests were conducted at the end of the pre-season macrocycle. Data on game indicators were record-ed during the 26 games played during the first and second rounds. The season was structured in eight macrocycles, one of which lasted five weeks and seven of which lasted four weeks. (Figure 1). The start of the preseason was during the month of September.”
- How was the sample selected? through an advertisement, a reference site, a phone call? Include eligibility and exclusion criteria and sources.
The following paragraph has been included to improve the explanation of the sample selection criteria.
“Convenience sampling was used, as it is a non-probabilistic and non-random sampling technique. This sample was selected according to the ease of access offered by the club and the availability of the female players to be part of the sample. The researchers selected this sample because the team was already participating in another research on the quantification of training and competition loads. The head coach was contacted first to find out his availability. After agreeing to participate in the study, permission was sought from the club.”
- The use of a figure to show the final recruitment of participants is recommended. It is a visual way to show the reader how the selection of participants has been made.
We appreciate your suggestion.
The authors consider that the recruitment of participants has been sufficiently explained in the paragraphs included in the Participants section
- How long did the study last? Indicate duration in months.
In the previous sections, the questions asked have been answered.
Please view the text again, as many changes have been made to substantially improve the manuscript.
2.3. Sample
- It is recommended to add the following sub-section: Eligibility criteria. The following can be added: “Only female basketball players were included in this study. Exclusion criteria for the study included: being involved in another type of sport at the same time as this study; not having a sufficient cognitive level to collaborate in the study or not being able to attend the established sessions".
The following section has been added under Participants.
“2.2.1. Eligibility criteria.
The following criteria were used to select the cases participating in the sample: i) Only female basketball players belonging to the club analyzed were included in this study; ii) To be considered a case study, the female player must have participated in at least 12 minutes per game [42]; iii) having played at least 1year professional category; iv) having performed the physical fitness tests during the pre-season; v) having com-pleted the two weeks of previous training, vi) not having suffered injuries in the first weeks of training.
Exclusion criteria were: i) participating less than 12 minutes of play in each match analyzed; ii) not having actively participated in the physical condition test, iii) being involved in another type of sport at the same time as in this study; iv) not having a sufficient cognitive level to collaborate in the study or not being able to attend the es-tablished sessions.”
The variables used were included separately for each of the variables used, for example: “As indicators of global strength, the Abalakov jump test [1] was performed. For the Abalakov jump test the participant also performed 3 countermovement jumps with 30 s rest between jumps, on a platform with an optical (infrared) data collection system (Optojump Next Microgate, Bolzano, Italy) to calculate the height of the Abalakov jump [2]. The player had to stand up and perform a 90° knee flexion followed by the fastest extension to reach the highest possible jump height. Of the three results, the best one was used for statistical analysis.
o 1. Wagner, H.; Sperl, B.; Bell, J.W.; Von Duvillard, S.P. Testing specific physical performance in male team handball players and the relationship to general tests in team sports. J. Strength Cond. Res. 2019, 33, 1056–1064.
o 2. Bosco, C.; Luhtanen, P.; Komi, P.V. A simple method for measurement of mechanical power in jumping. Eur. J. Appl. Physiol. Occup. Physiol. 1983.
The authors believe that their appreciation is very important. When analyzed in depth, we consider that its implementation in the manuscript would imply greatly expanding the text. Therefore, we have considered including a paragraph in which we refer readers to the primary source in which all the variables and the procedure for carrying out the tests are described.
" The precise description of all the physical fitness tests, as well as the process of obtain-ing the variables for each test, was carried out following the instructions defined by Mancha Triguero, Garcia-Rubio and Ibanez [30]"
For your information, we attach the procedure carried out.
As main indicators to evaluate the physical fitness, it was evaluated with the WIMUPRO device, different tests were chosen.
For the Abalakov test (Mancha-Triguero, García-Rubio, & Ibáñez, 2019) the participant also performed 3 countermovement jumps with 30 seconds rest between jumps (Heredia, Chirosa, Roldán, & Chirosa, 2009). For the Multijump test, the test begins by standing on the box with the height that belongs to the player, in order to perform a free fall and chain 5 maximum jumps in which the movement of the arms can help. The test will be performed twice by each subject, who will perform the test individually, with the recovery time between jumps being two minutes of passive recovery, since the test effort is similar to a 20-meter sprint (Wiewelhove, Raeder , Meyer, Kellmann, Pfeiffer, & Ferrauti, 2015).
For the repetition of effort test, it is proposed to carry out a Repeat Sprint Ability (RSA) test (Pion et al., 2015). The choice to evaluate this capacity is due to the fact that basketball is an interval sport in which maximum efforts are chained with incomplete recoveries (Jakovljevic et al., 2012). To do this, the athlete performs 5 14-meter sprints with a 30-second rest between sprints. For the agility test, the T test is selected (Fort-Vanmeerhaeghe, Montalvo, Latinjak, & Unnithan, 2016). The athlete has 2 opportunities to perform the test with a break between attempts of at least 2 minutes (Balsom, Seger, Sjödin, & Ekblom, 1992).
The Centripetal Force test selected is the Arc test (Mancha-Triguero, García-Rubio, & Ibáñez, 2019). The test consists of the athlete running at maximum speed on the three-point line painted on the field of play. The recovery time between tests will be 1 minute, since the effort is short and recovery is fast (Wiewelhove et al., 2015).
The selected aerobic capacity test is the SIG/AER Aerobic Test (Ibáñez, Sáenz-López, & Gutiérrez, 1995a). The choice of the test is due to its similarity with the competition itself, since it uses a greater number of formal elements of the game (playing field, ball and tactical-technical elements of the sport itself). It is a field test in which the athlete performs different technical-tactical actions of the sport for twelve minutes. It is the adaptation of the Cooper test to basketball.
Finally, to assess lactic anaerobic capacity, the SIG/ANA Anaerobic Test was chosen (Ibáñez, Sáenz-López, & Gutiérrez, 1995b). The selected test alternates the period of work and total passive recovery (ratio of work: rest is 1:1). Both tests, the SIG/ANA Anaerobic Test and the AGS/GRA test, retain the same structure. For the validation of the test, the results are correlated with the laboratory test, obtaining a high correlation (Ibáñez et al., 1995). The SIG/ANA Test (Ibáñez et al., 1995b) lasts ten minutes (including five minutes of activity and five minutes of recovery).
2.7. Statistic analysis
- Table 1. Indicate below the table the abbreviations used, such as df, S-W, etc.
Abbreviations are defined below the table
“S-W: Shapiro Wilk test; df: degree of freedom”
- Change Statistic for Statistical.
The proposed change has been made
- Line 190: some statistical equations are not cited, such as the Kruskal Wallis H test, please check the rest and include the relevant citations.
The use of the statistical test has been referenced.
[44]
Materials
- Check that the references included are those of the material used. These references refer only to WIMU PRO, and the study includes WIMU PRO, GARMIN ® and UWB [38-40].
Thank you for your comment.
The quotes are correct, as the WIMU PRO inertial device from Real Track Systems works synchronized with GARMIN heart rate bands. In order for inertial devices to work indoors, an ultra-wideband (UWB) radio frequency system must be installed. This entire system has been validated and published in the references provided.
RESULTS
- Line 258: Do not use the first person plural. Applicable to the rest of the document.
The suggestion is appreciated. The sentence has been corrected.
“The first group consisted of 49.1% of the sample and will be referred to as less important players (13.50 player minutes and 1.28 PIR). The second group consisted of 50.9% of the sample that will be referred to as important players (26.02 player minutes and 11.71 PIR).”
- Table 4a. The meaning of PIR is missing.
Thank you very much for your correction.
The meaning of the abbreviation PIR has been included in the tables, as well as in the study variables.
- To make it easier for the reader to understand the article, it is advisable to add a table with the socio-demographic characteristics of the sample. As it is done in table 2, it is first described and then mentioned. Where body composition variables (if measured) are also included.
The following information has been included at the beginning of the results section.
"Sociodemographic and body composition characteristics are presented in Table 2."
- Table 4a and 4b. When a table is split into two sheets, the sections must be put back in the header. Applicable to the rest of the manuscript.
The suggestion made is very relevant.
Tables 4a and 4b have been corrected.
DISCUSSION
- Discuss the generalisability (external validity) of the study results. The sample population must be considered.
All research carried out in the field of Social Sciences can commit or have the same problem. The generalizability of data when dealing with human persons with individual characteristics may be affected. In the same way, although the generalization of the results is not optimal, interesting conclusions can be drawn since the results presented by other players with physical, sporting and technical-tactical characteristics would not be very different. To this end, the following has been added to the document:
“This research was conducted in an ecological context, in which the research-ers did not alter or manipulate the participants' interventions. The validity of the data is very relevant, since there are few investigations that provide information on the relationship between physical and game performance in female players of basketball. The power of the results of this type of research is very great since it analyzes an entire professional team [41].”.
- Line 333: “These results are opposite with…for the intensity of the game or FP level”. By using reference Hulka, Cuberek, & Belka [53] you are comparing the results obtained in female players with junior male players, please try to compare them as far as possible with subjects with similar characteristics and games.
Your question is very interesting. However, the authors decided during the writing of the document to make this comparison. The main reason is that there is hardly any research carried out on women's basketball and with a professional sample. Therefore, after reviewing different documents, the most even results could be found in players in the last stage of training.
- The limitations of this research are not included in the Discussion section. Although it is included in the Conclusions, the limitation of the material used is not taken into account. It is recommended that other limiting aspects of this study be commented on. This research has recruited only female. This limitation has not been commented on.
Your question is very interesting.
For this reason, the authors believe that we must justify the selected sample. The main reason why only women are selected is focused on the fact that the players that make up a team are analyzed. In addition, the objective as a research team is to provide women's basketball with scientific evidence, since there is less research and evidence on training in this group of practitioners. Finally, the comparison with professional players was ruled out since the differences, both physical and technical-tactical, were very large.
The following limitations have been included.
“Another limitation of the replication of this research is that in order to obtain part of the information it is necessary to use high-cost technology, which is not always within the reach of all the teams. In spite of this, the battery of physical tests used allows the recording of different parameters without the need to use high-cost technological resources [30].
Finally, these results were obtained in a population of female basketball players, so their generalizability will be greater in this type of population.”
- The added figure under discussion is very complete, but does not include the Abalakov variable.
Thanks for your question. As for the figure, no test is mentioned, but instead talks about abilities or skills that athletes have. Within the jumping capacity there is both the Abalakov test and the multi-jump test.
- In general, some of the references included in this paragraph are not in line with the research population.
o Mendes FG, Lima AB, Christofoletti M, Quinaud RT, Collet C, Gonçalves CE, Carvalho HM. Multidimensional characteristics of young Brazilian volleyball players: A Bayesian multilevel analysis. PLoS One. 2021 Apr 30;16(4):e0250953. doi: 10.1371/journal.pone.0250953. PMID: 33930069; PMCID: PMC8087100.
Dear reviewer, the authors believe that this assessment does not correspond to our article, since this reference is not included in our manuscript.
The authors have tried to look for specific references to basketball.
CONCLUSIONES
- Line 468: The conclusion is a bit long; it is recommended to add these last two paragraphs in the discussion section and at the end.
Thank you very much for your comment.
The last two paragraphs of the conclusions have been moved to the end of the discussion section.
REFERENCES
- References follow the indicated style, but some of them are very old (more than five years).
- Some references are in Spanish
References have been updated.
It is true that some of them are written by Spanish-language authors, but they are quality sources specific to this research.

Reviewer 2 Report (New Reviewer)
STRUCTURE
The manuscript is generally well structured. However, in the materials and methods section there is a table of results that should be in the results section. (Table 1. Normality test results. On page 5).
TITLE AND ABSTRACT
In the title or abstract of the manuscript you should mention where the study was conducted and indicate what type of study it is. Whether it is an observational, analytical, descriptive study.
INTRODUCTION
The introduction is well written. It mentions scientific background and justification. It also states the specific objectives of the study.
MATERIAL AND METHODS
Design
They indicate that the proposed research is classified as a quasi-experimental study methodology. However, this is not sufficient because they should specify what this type of methodology is, or at least indicate whether there are any criteria for the selection of participants.
The word “design” is misspelled. They have written "desing". On page 2, last paragraph.
Participants
It should be explained that as this is a quasi-experimental methodology, the selection criteria is not random, therefore it would be important to indicate the reason or criteria for the selection of the players.
Why this team?
Why only 12 girls?
Which team is it?
In which country/city is the team located?
It would be interesting to have answers to such questions.
Sample
This section does not really understand why it is introduced when it is not mentioned again throughout the study just in the results section in line 258 “Distribution of players by game position based on their relevance in the game”. It is not understood why this sample has been introduced and why it recruits percentage data if the 12 participants are the ones with whom it is going to intervene. It does not seem relevant and, furthermore, there is a lack of information since it is mentioned that there is a series of cases, dividing some cases for the first round and others for the second. It also mentions an exclusion factor for the study when it should have been mentioned in the limitations section.
Statistic analysis
Here is a table of results that should be placed in the results section. On page 4, Table 1. Normality test results.
In this table are the abbreviations "df" which should be explained at the bottom of the table.
On the other hand, when it explains what it contains, there are inconsistencies between what it explains in the text and the results in the table. It would be interesting if those significant differences were reflected in the table itself, marked with an asterisk or some characteristic symbol to make it more visual.
Materials
Most women's or men's league clubs cannot afford to buy these material resources, which makes replicating the study and/or controlling internal and external loads with large, heterogeneous groups more complicated. Therefore, the use of this type of material hinders or limits the possibility of being able to replicate the study in other teams with smaller economic resources.
Instruments
On page 4, the acronym RSA should be described in full. In this case it is repeated sprint ability.
Procedure
It would be interesting if, in addition to mentioning the battery of tests used, you could provide a table, or an outline of which test was used to measure which variable. For example, did you use a contact platform, a mobile app or some other tool to measure jumps?
RESULTS
Table 1, which is placed in the statistical analysis section, is explained in section 3 of the results. This table shows some inconsistencies between what it explains and the data in the table. For example, Player Load (Anaerobic Capacity) present low or moderate effect size values. P value 0.734 is considered low or moderate, it could be that It refers to the distance which is 0.091 and even the HR Relative (%) with a p value 0.020, if we compare with the values you have mentioned. It is also noted that he is mixing the results of table 1 with table 2.
It makes no sense as you are talking about p values of .950 as is the height in the Abalakov test, for example, which is a high p value, not low or moderate.
As for table 3 it would be relevant that, when talking about the positions of the players, they indicate the number that compose them.
In addition, it would be interesting that those data that have a significant p-value, indicate it with an asterisk or some characteristic symbol.
Table 4a does not show Abalakov's test, but it does indicate the following:
“The Abalakov jump height variable (first and second round) and the duration time of the test and max. speed T Test without ball (first round) do not correlate with any Game Indicators variable.”
Although it indicates the time of the season when the trials take place (in pre-season to be exact), it does not indicate exact dates. Not knowing the origin of the basketball team, it is not even possible to guess the beginning of the season or the type of weather. It would be relevant to know the dates in order to know the time of the year and the type of weather, as this can affect performance regardless of whether the tests are carried out indoors or not. In this case it also does not indicate in which facilities the tests are carried out.
- Weiss, K., Valero, D., Villiger, E., Thuany, M., Cuk, I., Scheer, V., & Knechtle, B. (2022). Relationship between running performance and weather in elite marathoners competing in the New York City Marathon. Scientific Reports, 12(1), 21264. https://doi.org/10.1038/s41598-022-25901-z
DISCUSSION
This section lacks the limitations that have been encountered during the study. They are written as conclusions on line 475.
In line 316, Differences between PF according to game position, two studies are referenced, both of which are opposite as far as the results are concerned. It leaves an important doubt as to which of the two studies you have concluded is closer to your findings? (Fernández-Cortés, JA; Mandly, MG; García-Rubio, J; Ibáñez, SJ. Contribution of professional basketball players 550 according to the specific position and the competition phase. E-Bm com 2021, 17(3), 223-232. And Hulka, K; Cuberek, R; Bělka, J. Análisis de frecuencia cardíaca y movimiento de tiempo en los mejores jugadores 621 junior durante los partidos de baloncesto. Acta Gymnica 2013, 43(3), 27-35.)
In line 351, Relationship between GI and PF. On this occasion there are different results to those referenced with another study and the conclusion it gives is that this may be due to the level of competition assessed, however details are lacking as to what level the referenced study is.
REFERENCES
- References follow the indicated style
Author Response
Reviewer 2
Dear reviewer, the authors appreciate your work and the time spent in reviewing the manuscript. Below we detail and modify all the questions you have requested. The authors hope that they are to your liking.
Their contributions have improved the manuscript qualitatively
To facilitate their work, the authors have marked in red all the modifications made to the manuscript.
The authors made a considerable improvement to the article, responding point by point to each reviewer's suggestion/doubt.
The authors thank you for your kind words. It has been a great effort of work and time to collect and analyze the material to carry out this research.
I only add a few suggestions to the performance predict in the results, which makes the article innovative:
The authors appreciate your comments and the intention of the same.
Their contributions have substantially improved the document.
P.20, lines 257 to 265- Why did you use the PIR cluster and minutes played? As it stands, it makes no sense to answer the paper's objective ("...know the influence of the results of the Physical Fitness tests on the Game Indicators of professional basketball players"). The clusters formed (PIR with minutes played) are obvious (the longer the game time, the greater the chance of increasing the PIR) since PIR is a performance of the whole game and not relative per minute. I believe that it would be more interesting to answer the objective of the study if you try to verify the formation of clusters between PIR and the PF variables.
All their contributions are very pertinent and show a great knowledge of the problem under study.
First, a cluster was performed using minutes and PIR to identify the presence of more or less important female players, both globally and by specific positions.
A new Cluster analysis has been included, in which groups of players are identified by specific positions taking into account the physical fitness tests.
The authors did not initially include this second analysis in the study because the results made the manuscript too long.
We thank them for their suggestion and have included the analysis in the manuscript. This has resulted in the modification of the following sections:
Abstract:
“Finally, three groups of female players by playing position were identified according to their sport performance, PIR, high, medium, and low ratings, associated with physical performances. In all groups there are significant differences between playing positions, PIR and physical performances.”
Statistical analysis
“To graphically present the differences between the groups of players, the results were transformed into Z-scores and then normalized to a scale of 0-100 for visualization and comparison.”
Result
“Distribution of female players by playing position, their evaluation and physical fitness test results.
Subsequently, a more specific cluster analysis was performed in which we tried to identify groups of female players by playing positions including the PIR and the nineteen physical performance variables. Three good cluster models were identified for each specific position, in which the most important predictor was PIR and with equivalent sizes (Table 6).
|
Tabla 6. Physical and game performance profiles by playing positions |
|||||||||
|
|
Guard |
Forward |
Center |
||||||
|
|
PIR |
% |
p |
PIR |
% |
p |
PIR |
% |
p |
|
High PIR |
7.46 |
34.2 |
<.001 |
4.08 |
32.0 |
<.001 |
16.68 |
32.5 |
<.001 |
|
Medium PIR |
6.04 |
31.6 |
3.38 |
34.7 |
14.73 |
33.8 |
|||
|
Low PIR |
3.69 |
34.2 |
-0.32 |
33.3 |
3.38 |
33.8 |
|||
|
PIR: Performance Index Rating |
|||||||||
Specific profiles were found within each playing position for each group of female players according to their sport performance, PIR. Figure 1 shows the physical performance profiles for each player cluster by specific position.
“The female players with the highest rating during the game are the female players who obtained the best results in the ANA test (Distance, Explosive Dis-tance, Relative HR and Player Load), besides being the best in the Distance and Explosive Distance of the AER test, and Explosive Distance of the CFL test. On the other hand, the female players with the highest scores in the PIR obtained the best results in the ABK jump height, covered more Distance and Explosive Distance in the AER test, as well as had more Distance and Relative HR in the ANA test. Finally, the highest rated female players were the ones who had the best results in time and maximum speed in the T Test with and without the ball, they covered more Ex-plosive Distance and Explosive Distance in the AER test, as well as more Explosive Distance in the ANA test.”
Discusion
Relationship between physical and game performance profiles by specific positions.
Each playing position has a specific physical performance profile, which may be related to the roles coaches have assigned to them. The relationship between a higher score in the PIR of the female players with a better anaerobic capacity and a greater capacity to run distances in the aerobic test has been shown. The de-mands of the game require a high capacity to perform intermittent efforts, as well as to cover greater distances in these female players. Escudero-Tena, Rodríguez-Galan, García-Rubio and Ibáñez [22] identified the specific game actions that dif-ferentiate female players when winning and losing games. The 1-, 2-, and 3-point shots, as well as assists were identified as key [15,23]. Therefore, coaches should plan specific high-intensity tasks that allow the female players to outperform their opponents, in order to achieve a higher rating.
The female players who achieve higher ratings do so when they obtain better values in jump height, travel more distance and in an explosive way in the aerobic test. The ability to maintain continuous efforts at high intensity, moving from one side to the other in the front court to receive the ball, is very important for actions linked to a high PIR (shooting), as well as performing high intensity jumps, offensive and defensive rebounds. Total and defensive rebounds, as well as blocks made, free throws converted, and assists are the actions that differentiate for-wards when their teams win or lose [22,23]. These actions are linked to jumping height.
For their part, the female center obtained better PIR scores when they ob-tained better results in the T-Tests with and without the ball, as well as covering greater explosive distances in the aerobic and anaerobic tests. These female play-ers must make short cuts with continuous changes of direction in the spaces near the basket to gain position, which in turn will allow them to be effective in both the attack and defense phases. Their movements, in turn, must be at high intensity. The best center players when they win make more blocks, capture more defensive and total rebounds, score more 2-pointers, and assist their teammates [22,23]. These game indicators are linked to the ability to move quickly in tight spaces such as those used by these players.
Each type of female player requires specific physical training to improve the skills linked to the game actions that allow them to be more effective. Each coach must know which are the specific actions of effectiveness of the female players of his team, as these may vary depending on the style of play.
Please provide the p-value of the clusters.
The value of p has been included when comparing the clusters created.
The following sentence has been included:
"In all clusters the differences between the generated groups was significant (p<.001)."
Table 5- If PIR is the dependent variable and PF variables are the independent variables, why are minutes played as an independent variable? Minutes played should be used in the regression as a covariate (since it is a known PIR influencing variable) or be removed (because minutes players were not a PF variable).
This last point is very interesting and generated a wide debate among the authors of the paper.
Their recommendation has been followed and a new linear regression analysis has been performed, excluding minutes of play.
The results show fewer predictive models (from 6 to 3 models), with a lower explanatory capacity.
The complete results section has been changed, as well as the discussion.
The authors did not want to miss the opportunity to reflect the debate generated in the document, including this paragraph at the end of these results.
“Minutes of play were not included in the predictive model. There may be a relationship between a higher number of minutes played with a higher PIR. The more minutes played, the greater the possibility of performing positive actions that are included in the PIR calculation. However, the opposite may also occur. Also, there are cases in which female players play few minutes but obtain a high PIR. The regression model calculated that includes minutes played as an inde-pendent variable increases the predictive power of FP on PIR, in all playing posi-tions and during the two phases of the competition.”

Reviewer 3 Report (Previous Reviewer 1)
The authors made a considerable improvement to the article, responding point by point to each reviewer's suggestion/doubt.
I only add a few suggestions to the performance predict in the results, which makes the article innovative:
P.20, lines 257 to 265- Why did you use the PIR cluster and minutes played? As it stands, it makes no sense to answer the paper's objective ("...know the influence of the results of the Physical Fitness tests on the Game Indicators of professional basketball players"). The clusters formed (PIR with minutes played) are obvious (the longer the game time, the greater the chance of increasing the PIR) since PIR is a performance of the whole game and not relative per minute. I believe that it would be more interesting to answer the objective of the study if you try to verify the formation of clusters between PIR and the PF variables.
Please provide the p-value of the clusters.
Table 5- If PIR is the dependent variable and PF variables are the independent variables, why are minutes played as an independent variable? Minutes played should be used in the regression as a covariate (since it is a known PIR influencing variable) or be removed (because minutes players were not a PF variable).
Author Response
Reviewer 2
Dear reviewer, the authors appreciate your work and the time spent in reviewing the manuscript. Below we detail and modify all the questions you have requested. The authors hope that they are to your liking.
Their contributions have improved the manuscript qualitatively
To facilitate their work, the authors have marked in red all the modifications made to the manuscript.
The authors made a considerable improvement to the article, responding point by point to each reviewer's suggestion/doubt.
The authors thank you for your kind words. It has been a great effort of work and time to collect and analyze the material to carry out this research.
I only add a few suggestions to the performance predict in the results, which makes the article innovative:
The authors appreciate your comments and the intention of the same.
Their contributions have substantially improved the document.
P.20, lines 257 to 265- Why did you use the PIR cluster and minutes played? As it stands, it makes no sense to answer the paper's objective ("...know the influence of the results of the Physical Fitness tests on the Game Indicators of professional basketball players"). The clusters formed (PIR with minutes played) are obvious (the longer the game time, the greater the chance of increasing the PIR) since PIR is a performance of the whole game and not relative per minute. I believe that it would be more interesting to answer the objective of the study if you try to verify the formation of clusters between PIR and the PF variables.
All their contributions are very pertinent and show a great knowledge of the problem under study.
First, a cluster was performed using minutes and PIR to identify the presence of more or less important female players, both globally and by specific positions.
A new Cluster analysis has been included, in which groups of players are identified by specific positions taking into account the physical fitness tests.
The authors did not initially include this second analysis in the study because the results made the manuscript too long.
We thank them for their suggestion and have included the analysis in the manuscript. This has resulted in the modification of the following sections:
Abstract:
“Finally, three groups of female players by playing position were identified according to their sport performance, PIR, high, medium, and low ratings, associated with physical performances. In all groups there are significant differences between playing positions, PIR and physical performances.”
Statistical analysis
“To graphically present the differences between the groups of players, the results were transformed into Z-scores and then normalized to a scale of 0-100 for visualization and comparison.”
Result
“Distribution of female players by playing position, their evaluation and physical fitness test results.
Subsequently, a more specific cluster analysis was performed in which we tried to identify groups of female players by playing positions including the PIR and the nineteen physical performance variables. Three good cluster models were identified for each specific position, in which the most important predictor was PIR and with equivalent sizes (Table 6).
|
Tabla 6. Physical and game performance profiles by playing positions |
|||||||||
|
|
Guard |
Forward |
Center |
||||||
|
|
PIR |
% |
p |
PIR |
% |
p |
PIR |
% |
p |
|
High PIR |
7.46 |
34.2 |
<.001 |
4.08 |
32.0 |
<.001 |
16.68 |
32.5 |
<.001 |
|
Medium PIR |
6.04 |
31.6 |
3.38 |
34.7 |
14.73 |
33.8 |
|||
|
Low PIR |
3.69 |
34.2 |
-0.32 |
33.3 |
3.38 |
33.8 |
|||
|
PIR: Performance Index Rating |
|||||||||
Specific profiles were found within each playing position for each group of female players according to their sport performance, PIR. Figure 1 shows the physical performance profiles for each player cluster by specific position.
“The female players with the highest rating during the game are the female players who obtained the best results in the ANA test (Distance, Explosive Dis-tance, Relative HR and Player Load), besides being the best in the Distance and Explosive Distance of the AER test, and Explosive Distance of the CFL test. On the other hand, the female players with the highest scores in the PIR obtained the best results in the ABK jump height, covered more Distance and Explosive Distance in the AER test, as well as had more Distance and Relative HR in the ANA test. Finally, the highest rated female players were the ones who had the best results in time and maximum speed in the T Test with and without the ball, they covered more Ex-plosive Distance and Explosive Distance in the AER test, as well as more Explosive Distance in the ANA test.”
Discusion
Relationship between physical and game performance profiles by specific positions.
Each playing position has a specific physical performance profile, which may be related to the roles coaches have assigned to them. The relationship between a higher score in the PIR of the female players with a better anaerobic capacity and a greater capacity to run distances in the aerobic test has been shown. The de-mands of the game require a high capacity to perform intermittent efforts, as well as to cover greater distances in these female players. Escudero-Tena, Rodríguez-Galan, García-Rubio and Ibáñez [22] identified the specific game actions that dif-ferentiate female players when winning and losing games. The 1-, 2-, and 3-point shots, as well as assists were identified as key [15,23]. Therefore, coaches should plan specific high-intensity tasks that allow the female players to outperform their opponents, in order to achieve a higher rating.
The female players who achieve higher ratings do so when they obtain better values in jump height, travel more distance and in an explosive way in the aerobic test. The ability to maintain continuous efforts at high intensity, moving from one side to the other in the front court to receive the ball, is very important for actions linked to a high PIR (shooting), as well as performing high intensity jumps, offensive and defensive rebounds. Total and defensive rebounds, as well as blocks made, free throws converted, and assists are the actions that differentiate for-wards when their teams win or lose [22,23]. These actions are linked to jumping height.
For their part, the female center obtained better PIR scores when they ob-tained better results in the T-Tests with and without the ball, as well as covering greater explosive distances in the aerobic and anaerobic tests. These female play-ers must make short cuts with continuous changes of direction in the spaces near the basket to gain position, which in turn will allow them to be effective in both the attack and defense phases. Their movements, in turn, must be at high intensity. The best center players when they win make more blocks, capture more defensive and total rebounds, score more 2-pointers, and assist their teammates [22,23]. These game indicators are linked to the ability to move quickly in tight spaces such as those used by these players.
Each type of female player requires specific physical training to improve the skills linked to the game actions that allow them to be more effective. Each coach must know which are the specific actions of effectiveness of the female players of his team, as these may vary depending on the style of play.
Please provide the p-value of the clusters.
The value of p has been included when comparing the clusters created.
The following sentence has been included:
"In all clusters the differences between the generated groups was significant (p<.001)."
Table 5- If PIR is the dependent variable and PF variables are the independent variables, why are minutes played as an independent variable? Minutes played should be used in the regression as a covariate (since it is a known PIR influencing variable) or be removed (because minutes players were not a PF variable).
This last point is very interesting and generated a wide debate among the authors of the paper.
Their recommendation has been followed and a new linear regression analysis has been performed, excluding minutes of play.
The results show fewer predictive models (from 6 to 3 models), with a lower explanatory capacity.
The complete results section has been changed, as well as the discussion.
The authors did not want to miss the opportunity to reflect the debate generated in the document, including this paragraph at the end of these results.
“Minutes of play were not included in the predictive model. There may be a relationship between a higher number of minutes played with a higher PIR. The more minutes played, the greater the possibility of performing positive actions that are included in the PIR calculation. However, the opposite may also occur. Also, there are cases in which female players play few minutes but obtain a high PIR. The regression model calculated that includes minutes played as an inde-pendent variable increases the predictive power of FP on PIR, in all playing posi-tions and during the two phases of the competition.”

Round 2
Reviewer 1 Report (New Reviewer)
The authors have considered all the comments, and after each contribution, the modifications made are added appropriately.
The summary is a little longer and has more than 200 words. They should summarize it a little more. The rest of the article has improved quite a bit, as they have detailed the methodology and results much better.
It is also important that all tables and figures have the SAME CAPITAL LETTERS and SAME COLOR FORMAT.
The supplementary materials section should be deleted.
Unify the format of contributions, or all with . or with -.
In English, decimals are indicated by a period (.) not a comma (,). Check all tables.
Explanations of abbreviations in Figure 2 should be below the figure description.
If in table 5b the underlined is to be left, indicate what the green color means and what the orange color means.
What does "rho" mean? Explain in the table footer. Review all tables.
In Table 1, indicate the meaning of *. The design is a bit confusing. Review all tables.
Table 1 shows the ± sign next to the SD. Structure it so that it is in the middle. In the following tables mean and standard deviation appear without ±. Unify.
Author Response
REVIEWER 1
The authors have considered all the comments, and after each contribution, the modifications made are added appropriately.
Thank you so much for your words. The research team thanks you for positively assessing the work done.
Their contributions have improved the manuscript qualitatively
To facilitate their work, the authors have marked in red all the modifications made to the manuscript.
The summary is a little longer and has more than 200 words. They should summarize it a little more. The rest of the article has improved quite a bit, as they have detailed the methodology and results much better.
Thank you very much for your comment. The summary has been reduced to comply with the regulations of the journal. The reason it was too long is because the reviewers in previous reviews asked for more information.
It is also important that all tables and figures have the SAME CAPITAL LETTERS and SAME COLOR FORMAT.
Thank you very much for your comment. The research team has reviewed the document in its entirety and has made the necessary modifications to adapt the format. As for the table in figure 1, although it has a table format, it is a figure inserted as an image.
The supplementary materials section should be deleted.
Thank you very much for your comment, the supplementary material section has been deleted from the manuscript in its final version.
Unify the format of contributions, or all with . or with -.
Thank you very much for your comment. Regarding the use of “.” or “-“, the difference is due to the fact that one of the authors of the manuscript has ”–“ in his last name. However, the rest of the authors do not use the “–“because their scientific names do not require it.
In English, decimals are indicated by a period (.) not a comma (,). Check all tables.
Thank you very much for your comment that details the delicate and thorough revision that you have made of the document. The authors have reviewed the manuscript in its entirety, and it has been modified.
Explanations of abbreviations in Figure 2 should be below the figure description.
Thank you very much for your comment. The description of the variables has been added at the bottom of the figure.
If in table 5b the underlined is to be left, indicate what the green color means and what the orange color means.
Your question is very interesting. The description of the colors and their meaning is explained between lines 295-301. In addition, the meaning of the colors has been added to the description of the table. The authors hope that these explanations facilitate the understanding of the future reader.
What does "rho" mean? Explain in the table footer. Review all tables.
Thank you very much for your comment.
The meaning of "rho" is the result of Spearman's Correlation Coefficient, Rho (explained on line 250 of the manuscript). It is the statistical test for analyzing correlations between nonparametric variables. In the same way, your request is fulfilled, and the significance is added.
In Table 1, indicate the meaning of *. The design is a bit confusing. Review all tables.
The authors appreciate your comment. The reason for placing the "*" is due to a request from a reviewer of the manuscript. In addition, it is not something new, since it is a symbol widely used in the scientific field to highlight significant variables. In the same way, the meaning has been explained in the tables.
Table 1 shows the ± sign next to the SD. Structure it so that it is in the middle. In the following tables mean and standard deviation appear without ±. Unify.
The format has been unified. Thank you very much for your appreciation.

Reviewer 2 Report (New Reviewer)
No further comments.
Author Response
REVIEWER 2
No further comments.
The authors appreciate the work and dedication carried out, as well as the time spent in reviewing the manuscript. Your comments have helped to significantly improve the quality of the manuscript.

Reviewer 3 Report (Previous Reviewer 1)
The authors significantly changed the manuscript and managed to meet the main requests. The paper can be published.
Author Response
REVIEWER 3
The authors significantly changed the manuscript and managed to meet the main requests. The paper can be published.
The authors appreciate the work and dedication carried out, as well as the time spent reviewing the manuscript. Your comments have helped to significantly improve the quality of the manuscript.

This manuscript is a resubmission of an earlier submission. The following is a list of the peer review reports and author responses from that submission.
Round 1
Reviewer 1 Report
Thank you for the opportunity to review the manuscript. This is an interesting study whose objective is to assess whether the level of performance correlates with performance indicators. The important points of the study is that it was conducted with women who play basketball, both of which have been poorly studied. The study has an impact and can contribute to technical committees and coaches and propose reflections on the preparation of players. However, some issues deserve better described in the text and reflected by the authors.
Major review
Methods
-The PI data was from how many games?
Correlations
Was PI data controlled for minutes played in each game?
For example, was data entered for players who played only 10 minutes and players who played 30 minutes in the same game? If yes, the correlations presented in table 2 also must be controlled by the minutes player (need to create a new table?), because playing more minute increase the incidence of point, fault, running distance, Etc. Thus, such variables must be controlled for minutes played when they are correlated with PF. Therefore, the correlation between minutes played and PF must be presented as further analysis.
General cobments
No performance prediction was made from the PF and PI data. This is the main problem of the manuscript.
Results
P.2, Line 74- "into account" twice?
P.2, line 80- What does "ROI" mean?
P.2, Line 94- "Team members" refers to staff and players or only players. Please clarify throughout the paper.
P.3, Line 103- Is a comma missing after "PF"?
P.3, Line 21- Do not need to repeat the term "performance" after "PF". Please correct this throughout the paper.
Table 1- Please insert all unit measurements from the tests.
Table 1- . What does "ST" mean?
Table1. As was provided the P value, so do not need to indicate it with the *.
Table 1. ANOVA is significant in the "RSA Test" (and other variables), but post hoc did not identify the difference between player positions. Please clarify.
Table 1. ANOVA is not significant in "Expl. Distance" (P= 0.720), but post hoc did identify the difference between player positions. Please clarify.
Discussion
Although Figure 1 is interesting, it does not add anything to the discussion of the data presented in the results section. Authors need to use it better (use it to interpret your results) or remove it.
Author Response
Dear reviewer, all your suggestions and those of the other reviewers have been addressed.
The manuscript has been substantially improved.
Please find enclosed your individual response letter

Reviewer 2 Report
Dear authors,
I have thoroughly reviewed the manuscript titled: ‘’ Physical fitness as a predictor of performance in women's basketball players ‘’
This manuscript aims at influence of physical fitness on performance in basketball. As similar research of this type has been previously executed and the importance of PF is already highlighted, it is very important to point out the scientific and practical contributions of the manuscript.
There is a need to point out a few things in this manuscript which need to be further explained, change, or added. Please find attached the file with comments.

Author Response

(The authors gave the same response as above.)

Round 2
Reviewer 2 Report
I would like to thank the authors for the corrections made.